# Characterization of circulating breast cancer cells with tumorigenic and metastatic capacity

Claudia Koch[1,†], Andra Kuske[1,†], Simon A Joosse[1], Gökhan Yigit[2], George Sflomos[3], Sonja Thaler[4], Daniel J Smit[5], Stefan Werner[1], Kerstin Borgmann[6], Sebastian Gärtner[1], Parinaz Mossahebi Mohammadi[1], Laura Battista[3], Laure Cayrefourcq[7,8], Janine Altmüller[9], Gabriela Salinas-Riester[10], Kaamini Raithatha[10], Arne Zibat[2], Yvonne Goy[6], Leonie Ott[1], Kai Bartkowiak[1], Tuan Zea Tan[11], Qing Zhou[12], Michael R Speicher[12], Volkmar Müller[13], Tobias M Gorges[1], Manfred Jücker[5], Jean-Paul Thiery[14], Cathrin Brisken[3,15,‡], Sabine Riethdorf[1,‡], Catherine Alix-Panabières[7,8,‡] & Klaus Pantel[1,*]

## Abstract

Functional studies giving insight into the biology of circulating tumor cells (CTCs) remain scarce due to the low frequency of CTCs and lack of appropriate models. Here, we describe the characterization of a novel CTC-derived breast cancer cell line, designated CTC-ITB-01, established from a patient with metastatic estrogen receptor-positive (ER$^+$) breast cancer, resistant to endocrine therapy. CTC-ITB-01 remained ER$^+$ in culture, and copy number alteration (CNA) profiling showed high concordance between CTC-ITB-01 and CTCs originally present in the patient with cancer at the time point of blood draw. RNA-sequencing data indicate that CTC-ITB-01 has a predominantly epithelial expression signature. Primary tumor and metastasis formation in an intraductal PDX mouse model mirrored the clinical progression of ER$^+$ breast cancer. Downstream ER signaling was constitutively active in CTC-ITB-01 independent of ligand availability, and the CDK4/6 inhibitor Palbociclib strongly inhibited CTC-ITB-01 growth. Thus, we established a functional model that opens a new avenue to study CTC biology.

**Keywords** breast cancer; circulating tumor cells; functional studies; liquid biopsy; metastasis
**Subject Categories** Cancer; Stem Cells & Regenerative Medicine

## Introduction

Detection and characterization of circulating tumor cells (CTCs) have prognostic value in various tumor entities as demonstrated by several large clinical studies, e.g., for patients with breast and prostate cancer (Bidard *et al*, 2014; Goldkorn *et al*, 2014; Scher *et al*, 2015; Alix-Panabieres & Pantel, 2016). Moreover, these cells have the potential to be exploited as monitoring markers and might function as a blood-based biopsy guiding personalized treatment decisions (Keller & Pantel, 2019; Pantel & Alix-Panabieres, 2019). The perspective to accompany or even replace invasive tumor tissue biopsies in order to gain important diagnostically and therapeutically relevant information makes CTCs an essential contribution to

1  Department of Tumor Biology, Center of Experimental Medicine, University Medical Center Hamburg-Eppendorf, Hamburg, Germany
2  Institute of Human Genetics, University Medical Center Göttingen, Göttingen, Germany
3  ISREC – Swiss Institute for Experimental Cancer Research, School of Life Sciences, Ecole Polytechnique Fédérale de Lausanne (EPFL), Lausanne, Switzerland
4  European Centre for Angioscience (ECAS), Medical Faculty Mannheim, University of Heidelberg, Mannheim, Germany
5  Institute of Biochemistry and Signal Transduction, University Medical Center Hamburg-Eppendorf, Hamburg, Germany
6  Radiobiology& Experimental Radiooncology, University Medical Center Hamburg-Eppendorf, Hamburg, Germany
7  Laboratory of Rare Human Circulating Cells (LCCRH), University Medical Centre, Montpellier, France
8  Montpellier University, Montpellier, France
9  Cologne Center for Genomics, University of Cologne, Cologne, Germany
10  NGS Integrative Genomics Core Unit, Institute of Human Genetics, University Medical Center Göttingen, Göttingen, Germany
11  Cancer Science Institute of Singapore, National University of Singapore, Singapore City, Singapore
12  Institute of Human Genetics, Diagnostic and Research Center for Molecular BioMedicine, Medical University of Graz, Graz, Austria
13  Department of Gynecology, University Medical Center Hamburg-Eppendorf, Hamburg, Germany
14  INSERM Unit 1186, Comprehensive Cancer Center, Institut Gustave Roussy, Villejuif, France
15  Breast Cancer Now Research Centre, Institute of Cancer Research, London, UK
*Corresponding author. Tel: +49 40 741053503; Fax: 49 40 7410-55379; E-mail: pantel@uke.de
†These authors contributed equally to this work as first authors
‡These authors contributed equally to this work as senior authors

non-invasive "*real-time liquid biopsies*" (Pantel & Alix-Panabieres, 2010; Bardelli & Pantel, 2017).

In spite of an enormous progress in the development of approaches for the detection and molecular characterization of CTCs up to the single cell level (Joosse *et al*, 2012; Alix-Panabieres & Pantel, 2014a,b; Pantel & Alix-Panabieres, 2019), information on the functional properties of CTCs is still limited due to the very low concentrations of these cells in the peripheral blood of patients with cancer (Alix-Panabieres & Pantel, 2014a).

Not all CTCs possess the potential to extravasate at distant sites and grow out to form a novel metastatic lesion (Wicha & Hayes, 2011). A plethora of different factors play into the survival of these CTCs in the blood stream and their capacity to extravasate and metastasize (Strilic & Offermanns, 2017; Giuliano *et al*, 2018), including the hemodynamic forces within the circulation (Follain *et al*, 2018) and genomic make-up of the tumor cells (Joosse & Pantel, 2016; Gkountela *et al*, 2019). Experimental models indicate that only few tumor cells are viable, survive shear forces within the blood flow, evade the immune system as well as systemic therapies, reach distant organs, and eventually have the potential to form an overt metastasis (Chambers *et al*, 2002, Y. Kang & Pantel, 2013). Adaptation to a new microenvironment and proliferation of a single tumor cell or a CTC cluster in a distant site requires highly specialized traits, most of which are largely unknown. In order to understand these underlying mechanisms of the metastatic cascade, functional characterization of viable CTCs capable of forming distant metastasis is required.

A prerequisite for these analyses were therefore the recent advances in the ability to culture CTCs *in vitro* (Zhang *et al*, 2013; Yu *et al*, 2014; Cayrefourcq *et al*, 2015) or expand the CTC pool *in vivo* using xenografts (Baccelli *et al*, 2013; Hodgkinson *et al*, 2014; Carter *et al*, 2017). However, to our knowledge, none of these studies compared the characteristics of the original CTCs captured from the patients with cancer directly to the CTC line. Besides unraveling the biology of CTCs in patients with cancer, these studies allow testing of the unknown activity of cancer drugs against CTCs. In this context, the low number of estrogen receptor-positive (ER$^+$) breast cancer cell lines presently available is disturbing, since 70–80% of patients with breast cancer harbor ER$^+$ tumors and ER is the primary target of endocrine therapies in breast cancer (Pan *et al*, 2017).

Here, we report the establishment and in-depth characterization of an ER$^+$ breast CTC line with unique properties. Comparison of CTCs *in situ* before cell culture with the CTC line indicates that it mirrors the situation in ER$^+$ breast cancer patients and therefore provides novel insights into the biology and drug response of patient-derived CTCs in the most common breast cancer subtype.

# Results

## Patient characteristics

The patient with MBC the CTC-ITB-01 cell line derived from presented with a bilateral mammary carcinoma, lymph node (LN) metastases, and bone metastases (BM) at age 75, 2 years prior to blood collection for CTC analysis and begin of cultivation (Fig 1A). Histopathological analysis of biopsies performed for both breast tumors revealed a well-differentiated (G1) invasive lobular carcinoma (ILC) of the left breast and a well-differentiated (G1) invasive ductal carcinoma (IDC) of the right breast (Table EV1). E-cadherin re-staining of the tissue biopsies from both tumors revealed that the tumor described as lobular showed a sub-fraction of E-cadherin$^+$ (Fig EV1A) besides E-cadherin$^-$ tumor cells (Fig EV1B). This could point toward a ducto-lobular histopathology. Both the primary right ductal tumor (Fig EV1C) and the vaginal metastasis (Fig EV1D) contained strongly E-cadherin$^+$ tumor cells.

Both primary tumors were classified as estrogen receptor-positive (ER$^+$ in ≥ 80% of nuclei), progesterone receptor positive (PR$^+$ in ≥ 80% of nuclei), ERBB2 negative (ERBB2$^-$), and presented with a low Ki67$^+$ cell fraction of 5%. At time of blood collection, additional metastases in the spleen, liver, and vagina had been diagnosed. An overview of the disease progression and treatment scheme of this patient can be found in Fig 1A and in Appendix Supplementary Methods.

## Establishment of a CTC-derived breast cancer cell line

The peripheral blood sample of this patient with metastatic breast cancer (MBC) was screened for CTCs using the CellSearch$^®$ System, resulting in a tumor cell count of 1,547 CTCs per ml of blood (total of 7.5 ml). In parallel, blood of the same patient was processed for cell culture. This led to the establishment of a permanent cancer cell line, designated CTC-ITB-01. The original CTCs at blood draw were comprised of single cells of various shapes and diameters (Fig 1B, e.g., panels 1–3), as well as approximately 700 small cell clusters of CTCs (Fig 1B, panels 5–6). CTCs showed negative or very weak (Fig 1B, panels 4 and 8) immunostaining for ERBB2. CTC-ITB-01

**Figure 1. CTC cell line establishment from peripheral blood of an mBCa patient.**

A Scheme of the breast cancer patient's clinical status and therapies. Course of disease progression (blue) and treatment scheme (green) of the patient giving rise to CTC-ITB-01 are indicated. Timeline of progression and treatment indicated in years and months (mo). Drugs were administered at standard dosage according to indicated pattern. The red star represents the time point of blood sample collection. More detailed information is available the in Appendix Supplementary Methods.

B Representative pictures of different CTCs from the initial CellSearch$^®$ analysis of the metastatic breast cancer patient who gave rise to the breast CTC line. The detected tumor cells display clear keratin and DAPI staining, CD45 negativity as well as lack of, or very weak (4, 8), ERBB2 expression. Cells of small (about 5 μm in diameter, 1, 2) and large size (larger than 10 μm in diameter, 3) were detected. While some CTCs displayed dot-like perinuclear keratin signals (1, 2), the majority showed diffuse keratin staining. Additionally, CTC clusters of more than 4 cells were present (5, 6). Some CTCs showed multiple/lobed nuclei (7, 8).

C Bright field images of CTC-ITB-01 cells growing adherently.

D Bright field images of CTC-ITB-01 cells growing non-adherently (relation between adherent and non-adherent cells: 80/20%).

Data information: White scale bars represent 10 μm. Black scale bars represent 40 μm.

Source data are available online for this figure.

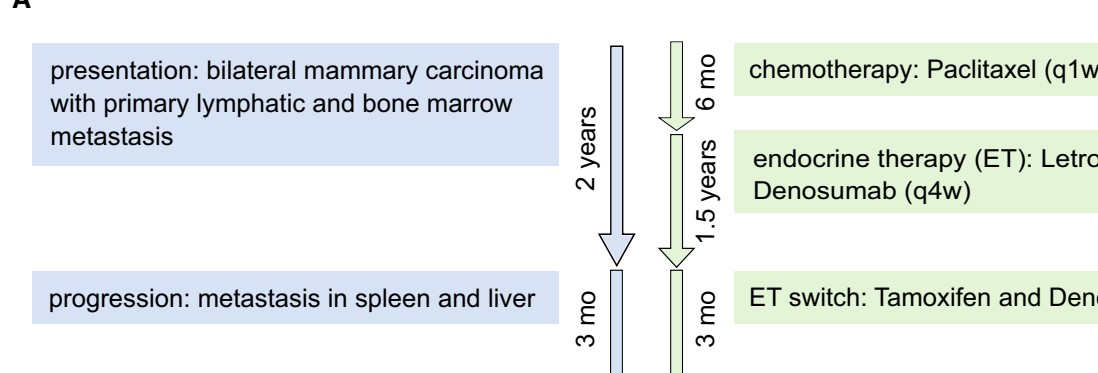

Figure 1.

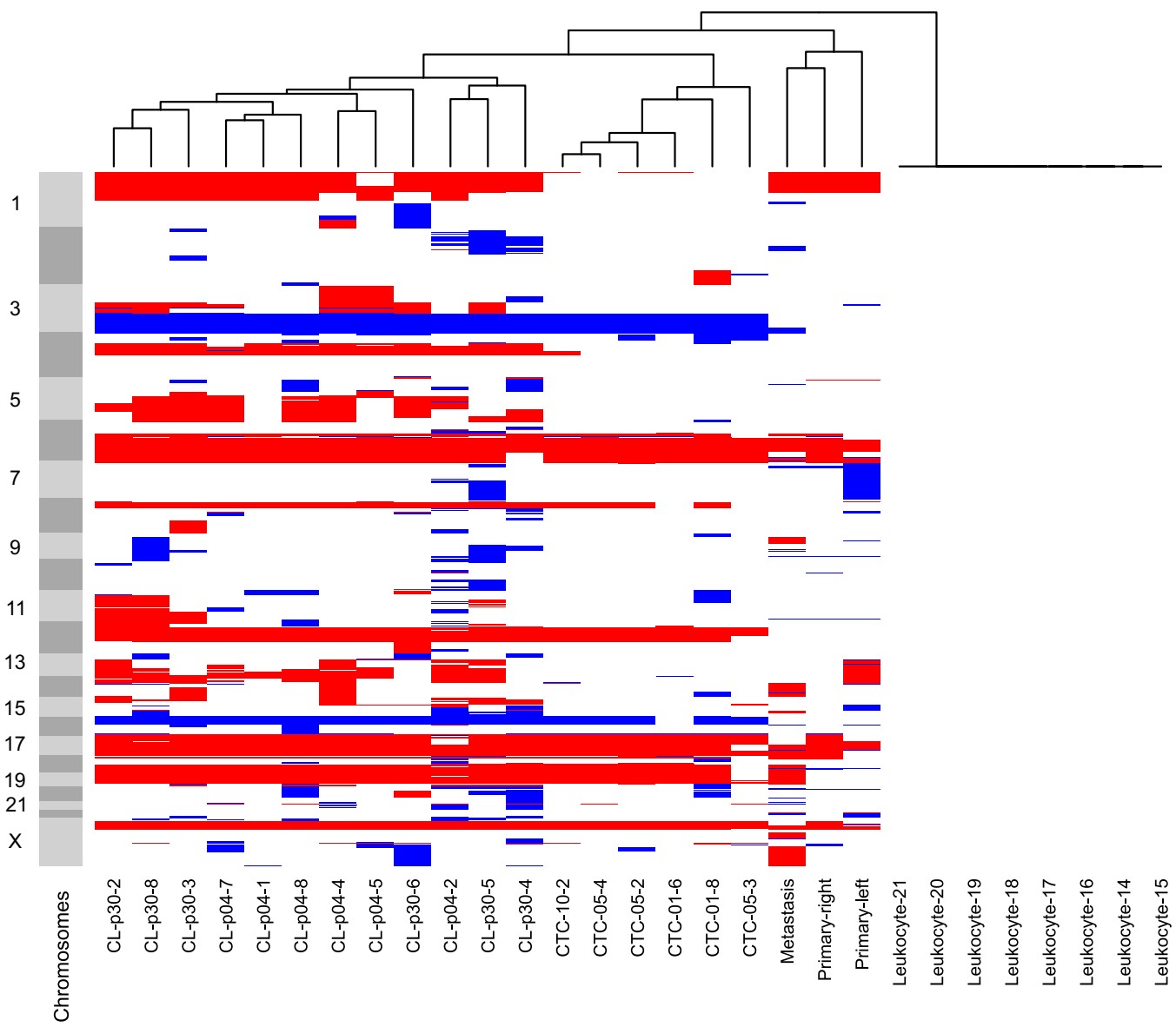

**Figure 2. Hierarchical clustering analysis of CTC-ITB-01.**

Unsupervised hierarchical clustering (Ward linkage with Euclidean distance) of CNA profiles generated from patient primary tumors, the vaginal metastasis, isolated single and pooled CTCs from the primary blood collection and CTC-ITB-01 (early and later passage) and leukocytes. "CL" represents the cell line, followed by the passage analyzed (p04/p30). "CTC" marks circulating tumor cells isolated from the initial blood draw, followed by 01/05/10 indicating single (01) or pooled cells (5 or 10, respectively). Light and dark gray bars to the left signify chromosomes, starting from the top with chromosome 1. Red color represents chromosome loss, blue represents gain. All copy number aberrations were calculated using the patient's own germline DNA extracted from leukocytes as reference to eliminate copy number variations (CNVs).

has now been successfully cultured for more than 4 years and cells grow in a mixed epithelial–mesenchymal morphology (Fig 1C) as well as in adherent (Fig 1C) and non-adherent fractions (Fig 1D).

## Genomic characteristics of CTC-ITB-01 cells

Remarkably, a considerable number of the CTC cell line cells carried multiple and lobed nuclei, probably due to abnormal cytokinesis (Fig EV2A). Those cells were also detectable in the original Cell-Search analysis (Fig 1B, panels 7–8), indicating that this is not an artificial effect originating in cell culture. Giant cancer cells carrying multiple nuclei have recently been associated with metastasis and

disease relapse (Mirzayans *et al*, 2018). Karyotyping of the cell line showed a broad range of chromosomes per cell (32–110), resulting in a mean of 70.7 (s = 17.66) chromosomes (Fig EV2B and C).

Whole-genome next-generation sequencing of single cells of the CTC-ITB-01 line revealed a wide spectrum of copy number alterations (CNA) including loss of large areas of chromosome 16q (Fig 2), typical for ER$^+$ breast cancers (Horlings *et al*, 2010). Hierarchical clustering analysis showed that the primary CTCs within the patient's blood and the resulting CTC-ITB-01 line cluster separately to the original primary tumors (left and right breast) and the vaginal metastasis, indicating tumor evolution during the course of the disease (Fig 2).

Overall, the highest concordance in CNA profiles was detected between the arbitrarily collected sample of original CTCs analyzed and the CTC-ITB-01 line (Fig 2), providing strong evidence that the CTC line derived from a subpopulation of CTCs in circulation at time point of blood draw. We furthermore directly compared the CNA profiles of CTC-ITB-01 cells to those of the primary CTCs present in the patient's blood (Fig 3A). Common aberrations include gain of chromosomes 3q and 15q as well as loss of chromosomes 6q, 12p, 16q, 17p, 18, and 22q (Fig 3A). Additionally, CTC-ITB-01 remains stable in its CNA profile during culture, shown by direct comparison of an early (04) with a late passage (30) of the line (Fig 3B). The fact that early and late passages of the line cluster together (Fig 2) represents another strong indicator of the CTC line's stability. Loss of chromosome 1p represents the largest significant difference measured between the early and late passages (Fig 3B).

Whole exome sequencing (WES) of CTC-ITB-01 cells, the two primary tumors, and the distant vaginal metastasis was performed and data were analyzed for mutations in common cancer-related genes (Table 1). We detected a one base pair deletion in MAP3K1 (c.2782delT; p.S928Lfs*9) leading to an amino acid frameshift, thereby causing protein truncation and loss-of-function of the protein (Pham et al, 2013; Avivar-Valderas et al, 2018) in all four samples. Further, the two primary tumors, the vaginal metastasis and the CTC cell line also shared a mutation in the gene region encoding the kinase domain of the PIK3CA protein (c.3140A>G; p.H1047R, Table 1), a somatic hot spot mutation site in lobular and ductal breast cancer that has been associated with increased enzymatic activity of PIK3CA (Kang et al, 2005; Stemke-Hale et al, 2008). Besides shared variations with the primary tumors, CTC-ITB-01 and the vaginal metastasis exhibited an additional, less frequent PIK3CA mutation (c.1252G>A; p.E418K, Table 1), located in the region encoding the C2 calcium/lipid-binding domain (Saal et al, 2005; Stemke-Hale et al, 2008). Moreover, we identified a genomic aberration of the NF1 gene that was shared between the CTC-ITB-01 cell line, the lobular tumor and the vaginal metastasis, but also "private" CDH1 (c.2466delC; p.T823Qfs*23) and TP53 (c.1024C>T; p.R342*) gene mutations in the vaginal metastasis not observed in the primary tumors or the CTC line (Table 1).

We furthermore checked for genes typically associated with hereditary breast cancer predisposition (e.g., BRCA1/2, TP53, PTEN, STK11 or CHEK2), but found no mutations in the primary tumors. Also, no mutations were detected for other described breast cancer-associated genes (e.g., RAD51, RAD50, p27, ESR1/2, ERBB2, ERBB3, AKT1, CCND1, FGFR1, MYC, and RB1) in any of the tissues or the CTC cell line.

Another specific variation of CTC-ITB-01 is a homozygous, mutation in the TP53 tumor suppressor gene (c.853G>A; p.E285K)

(Table 1). The TP53 gene sequence is commonly altered in breast cancer (Bertheau et al, 2013; Dumay et al, 2013), and this point mutation represents an established pathogenic variant (Fig EV3A–C) (Xu et al, 1997; Oh et al, 2000). Additionally, immunostaining of the p53 protein showed a strong accumulation of this tumor suppressor in the nucleus of CTC-ITB-01 cells (Fig EV3D).

Furthermore, the CTC-ITB-01 line carries a "private" variant in the CDH1 gene (c.1204G>A; p.D402N, Table EV2) that differs from the CDH1 mutation of the lobular primary tumor (c.1792C>T; p.R598*, Table 1). While the p.R598* mutation that has been previously detected in ER$^+$ BCa tissue (Ma et al, 2017) represents a nonsense substitution leading to loss-of-function of the CDH1 protein, the CDH1 mutation of the CTC cell line is a relatively uncharacterized missense mutation in E-cadherin's extracellular domain 3 (EC3) which is necessary for homophilic adhesion (Shiraishi et al, 2005). Apart from CDH1, we also checked for variations in other genes distinguishing the different histopathological subtypes of the two primary tumors (i.e., ILC and IDC), such as GATA3, FOXA1, and TBX3; however, no additional mutations of these genes were detected in the CTC line. Further analysis of WES data revealed additional variants in common cancer-related genes involved in DNA replication and DNA damage (e.g., ATM, CDKN1A), cell polarization (SCRIB), proliferation (RNASEL), VEGF expression (MAP3K1, MAP3K6), and in genes encoding growth factors (FGF2) (Table EV2). These variants were predicted to impair protein function by different in silico prediction tools; however, experimental data on the functional relevance of these variants are still inconsistent.

Finally, the CTC-ITB-01 line displayed a high degree of loss of heterozygosity (LOH) as assessed by runs of homozygosity (ROH) within the exome. In comparison with the primary tumors which showed homozygosity within < 90 Mb of the genome, 601 Mbs of homozygous sequences were detected for the breast CTC line, potentially indicating heterozygous mutations that became homozygous and contributed to tumor evolution. The 15 largest regions of LOH found solely in CTC-ITB-01 are listed in Table EV3 and are also discernible in Fig 2.

## Expression signature of CTC-ITB-01 cells

CTC-ITB-01 cells were first characterized for ER and ERBB2 protein expression (receptors relevant to the major breast cancer subtypes) by Western blot (Fig 4A). ER expression in CTC-ITB-01 cells was strong, albeit less pronounced than in MCF-7 cells, the standard model for ER$^+$ breast cancer cells (Fig 4A). ER positivity was confirmed via immunocytochemistry (Fig 4B). Consistent with the lack of ERBB2 amplification (Appendix Fig S1A), ERBB2 expression was

**Figure 3. CNA profile analysis of CTC-ITB-01.**

Direct comparisons of CNA profiles. Percentages of gene copy number gains and losses across the chromosomes for different sample types. Copy number gains are displayed in green, losses in red.

A CNAs generated from CTCs from the primary blood collection (upper panel) and CTC-ITB-01, indicated as "CL" (medium panel). Significant distinctions between both samples were calculated using Fisher's exact test (Joosse et al, 2018) and are indicated in lowest panel (blue). Increased blue color intensity represents lower P-value and therefore higher statistical significance.

B CNA comparison between early (p04) and late (p30) CTC-ITB-01 passages (upper and medium panel). Significant distinctions between both samples were calculated using Fisher's exact test (Joosse et al, 2018) and are indicated in lowest panel (in blue). Increased blue color intensity represents lower P-value and therefore higher statistical significance.

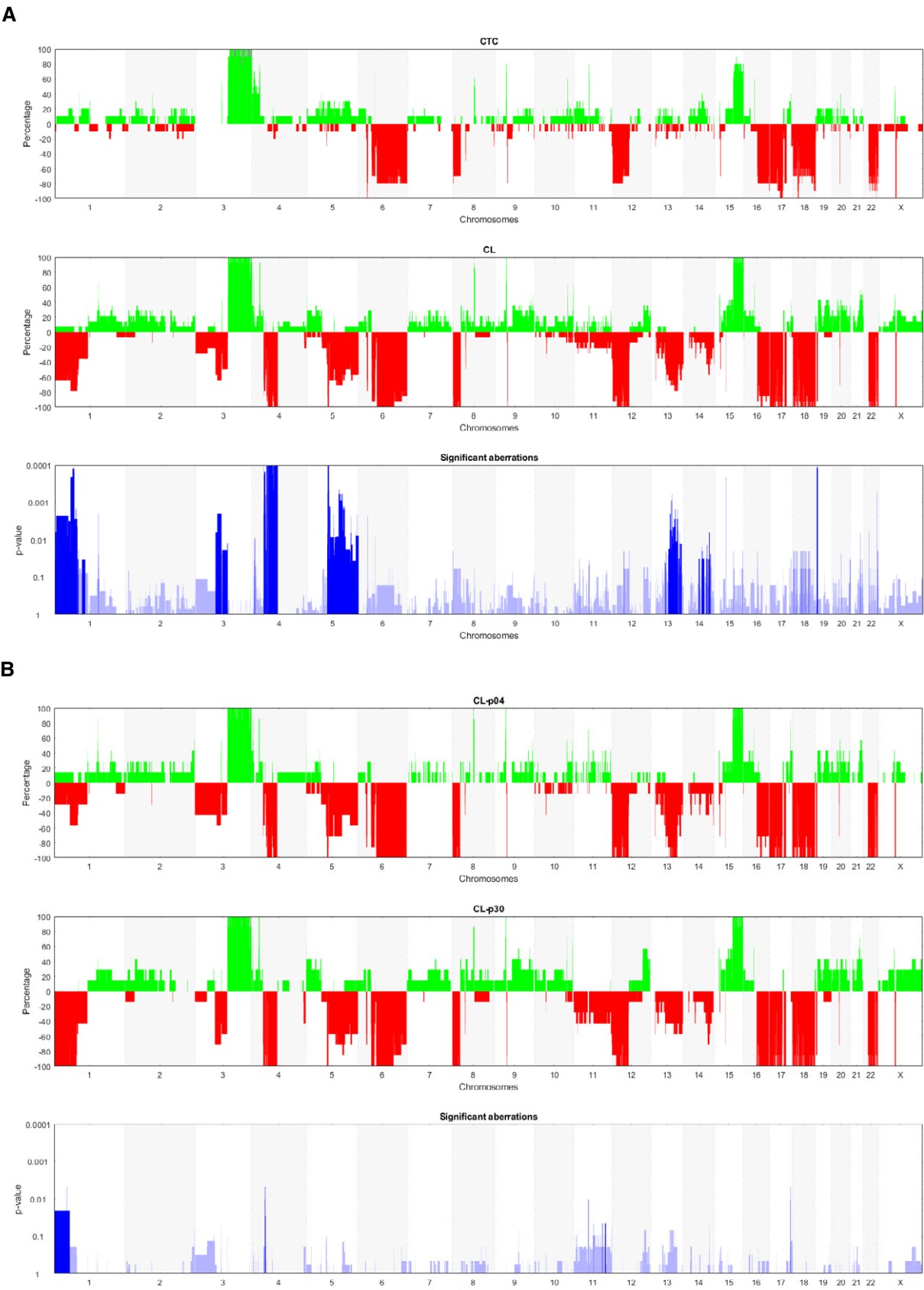

**Figure 3.**

**Table 1. Functionally relevant mutations in CTC-ITB-01, vaginal metastasis, and primary tumors identified by whole-exome sequencing.**

| Gene name | CTC-ITB-01 | Vaginal metastasis | Left primary tumor | Right primary tumor |
|---|---|---|---|---|
| *CDH1* | – | – | c.1792C>T; p.R598* (13%) | – |
| | – | c.2466delC; p.T823Qfs*23 (37%) | – | – |
| *MAP3K1* | c.2782delT; p.S928Lfs*9 (41%) | c.2782delT; p.S928Lfs*9 (15%) | c.2782delT; p.S928Lfs*9 (22%) | c.2782delT; p.S928Lfs*9 (16%) |
| *MAP3K6* | c.2837C>T; p.P946L (30%) | c.2837C>T; p.P946L (40%) | c.2837C>T; p.P946L (30%) | c.2837C>T; p.P946L (45%) |
| *NF1* | c.4528_4529insG; p.L1510Rfs*20 (90%) | c.4528_4529insG; p.L1510Rfs*20 (37%) | c.4528_4529insG; p.L1510Rfs*20 (4%) | – |
| *PIK3CA* | c.1252G>A; p.E418K (26%) | c.1252G>A; p.E418K (33%) | – | – |
| | c.3140A>G; p.H1047R (74%) | c.3140A>G; p.H1047R (40%) | c.3140A>G; p.H1047R (21%) | c.3140A>G; p.H1047R (23%) |
| *TP53* | c.853G>A; p.E285K (92%) | – | – | – |
| | – | c.1024C>T; p.R342*(39%) | – | – |

All four samples were analyzed for rare (MAF, minor allele frequency, <1%), functionally relevant variants with an allele frequency of 10% of reads and higher in at least one of the samples. 219 cancer-associated genes curated from the COSMIC, HGMD, and OMIM databases were analyzed, and 9 pathogenic mutations in 6 genes were identified. Gene symbols were used as approved by the HGNC, and location of variants on cDNA and protein (one letter code) level, and allele frequency is shown. del, deletion; ins, insertion; *, stop codon; fs, frameshift.

weak in CTC-ITB-01 cells (Fig 4A, Appendix Fig S1B). This phenotype mirrors the ER positivity and ERBB2 negativity of both primary tumors (Table EV1). We furthermore used RNA sequencing to perform expression-based subtyping of the CTC cell line using the PAM50 and scmod2 classifiers (Wirapati *et al*, 2008; Parker *et al*, 2009). Pooled cell line cells, as well as the adherent and non-adherent cell fraction, were classified as the breast cancer luminal B subtype (Fig 4C and D).

CTC-ITB-01 cells showed reduced expression of epithelial intermediate filaments K8 and K18 as compared to MCF-7 (Fig 4A), while the mesenchymal markers N-cadherin and vimentin were not expressed (Fig 4A). We further investigated the expression of breast cancer stemness markers. Aldehyde dehydrogenase activity (termed ALDH$^+$) and CD44$^+$/CD24$^-$ mark two largely non-overlapping populations of cancer stem cells, which have epithelial-like and mesenchymal-like phenotypes, respectively (Liu *et al*, 2014). CTC-ITB-01 cells were CD44$^-$/CD24$^+$ (Fig 4A) and a large proportion (77.92%) were ALDH$^+$ as revealed by flow cytometric analysis (Fig 4E) and confirmed by ICC (Fig EV4).

The capacity of CTC-ITB-01 cells to secrete specific proteins of interest was assessed via functional fluoro-EPISPOT assays.

Besides keratin 19, recently described as CTC marker for breast cancer and other epithelial malignancies (Alix-Panabieres & Pantel, 2014a), viable CTC-ITB-01 cells actively secreted vascular endothelial growth factor (VEGF) known to induce tumor angiogenesis in patients with cancer (Saharinen *et al*, 2011) (Appendix Fig S2). While this secretion is comparable to that of the colon cancer CTC cell line CTC-MCC-41, CTC-ITB-01 does not secrete OPG (osteoprotegerin), EGFR (epidermal growth factor receptor) or FGF2 (fibroblast growth factor-2) characteristic for CTC-MCC-41 (Appendix Fig S2) (Cayrefourcq *et al*, 2015).

To find out whether stem cell pathways might be involved in the stem cell-like behavior of CTC-ITB-01, we tested a selection of index proteins by Western blot analysis. Interestingly, the expression level of NUMB indicative of activation of the NUMB pathway was strikingly increased compared with MCF-7, while NOTCH1 and NOTCH3 as components of the NOTCH pathway are rather weaker expressed in CTC-ITB-01 than in MCF-7. The levels of Cleaved NOTCH-1 expressed in CTC-ITB-01 and MCF-7 were similar (Fig EV5). Our results imply a participation of these pathways in the regulation of stem cell features in the CTC-ITB-01 cell line (Bocci *et al*, 2017; Saha *et al*, 2017).

**Figure 4. CTC-ITB-01 phenotype in culture.**

A Western blot analysis of selected protein markers, including ERα, EGFR, ERBB2, EpCAM, K18, K19, K8, E-cadherin, N-cadherin, vimentin, CD44, CD24, SNAIL, SLUG, TWIST1, and α-tubulin (as a loading control) (*n* = 3 replicates). CTC-ITB-01 was compared to more mesenchymal ER$^-$ Hs578t and epithelial ER$^+$ MCF-7 breast cancer cell lines.

B ICC staining of CTC-ITB-01 for pan-keratin (orange), ER (green), and DAPI (blue). The scale bar corresponds to 20 μm. Two representative panels are shown for CTC-ITB-01 (1) and (2). MCF-7 cells are depicted as reference cell line.

C PAM50 classifier results showing probabilities of pooled CTC-ITB-01 cells matching specific molecular breast cancer subtypes. Starting from lowest probability, CTC-ITB-01 was classified as 1.02% (s = 2.5%) normal-like, 3.11% (s = 6.6%) Basal-like, 13.11% (s = 9.3%) ERBB2-positive, 16.77% (s = 23.7%) luminal A, and 65.22% (s = 16.6%) luminal B breast cancer subtype. Data were generated from *n* = 3 replicates.

D PAM50 classifier results showing probabilities of the non-adherent and adherent CTC-ITB-01 fractions matching molecular subtypes. Selected bars are not visible due to extremely low probability (close to zero). Both fractions show greatest alignment with a luminal B subtype. Starting from lowest probability, the adherent fraction of CTC-ITB-01 was classified as 0% (± 0%) normal-like, 6.22% (± 9.0%) Basal-like, 17.85% (± 8.6%) ERBB2-positive, 0% (s = 0%) luminal A, and 75.92% (s = 0.7%) luminal B breast cancer subtype. The suspension cell fraction of CTC-ITB-01 was classified as 2.04% (s = 3.5%) normal-like, 0% (s = 0%) Basal-like, 17.32% (s = 9.7%) ERBB2-positive, 33.5% (s = 23.8%) luminal A, and 68.67% (s = 18.6%) luminal B breast cancer subtype, respectively. Data were generated from *n* = 3 replicates.

E ALDH activity measurement on viable cells via flow cytometry using the ALDEFLUOR™ assay. CTC-ITB-01 is compared to A549 as recommended control cell line. An internal (DEAB) control is also depicted. 77.92% of the gated CTC-ITB-01 population are ALDH$^+$.

Source data are available online for this figure.

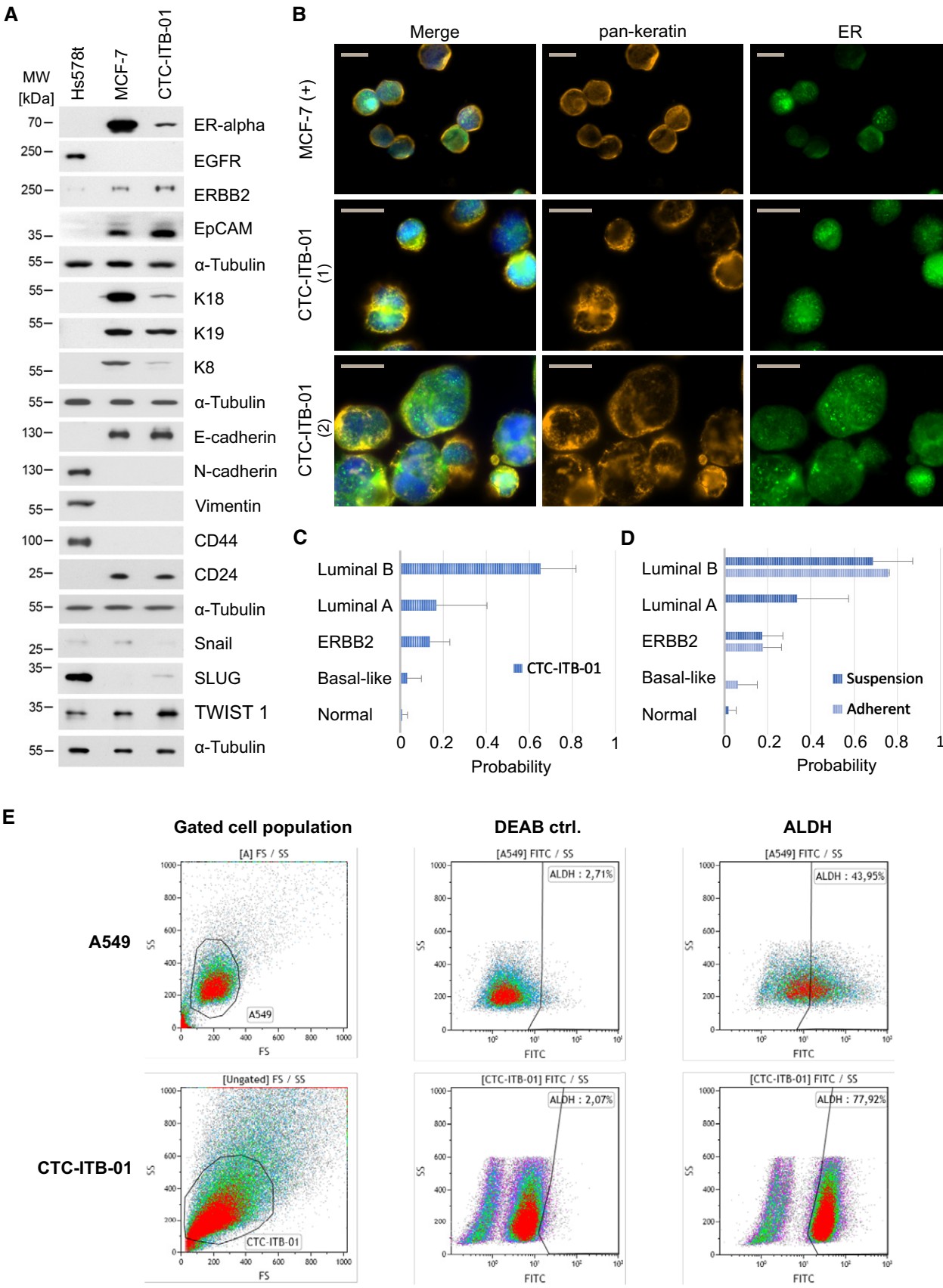

**Figure 4.**

**Epithelial–mesenchymal plasticity of CTC-ITB-01 cells**

In the initial blood sample of the patient with breast cancer, the high number of more than 1000 CTCs per ml of blood was detected with the FDA-cleared CellSearch® system. CellSearch® uses magnetic particles coupled to antibodies against EpCAM to enrich fixed CTCs and subsequently identifies single CTCs by immunostaining of epithelial keratins. Thus, we conclude that this patient harbored

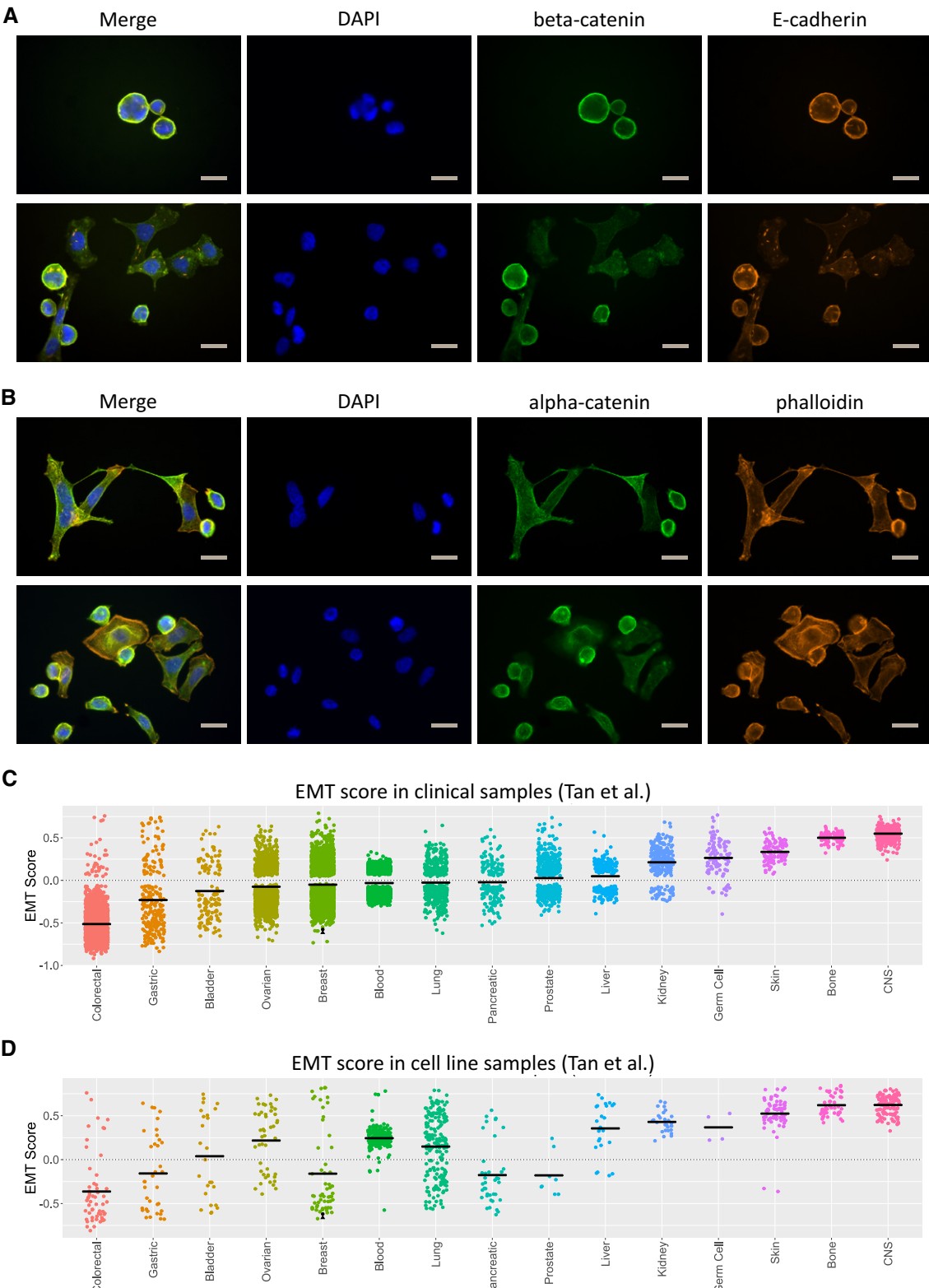

**Figure 5.**

**Figure 5. Epithelial adhesion and EMT analysis of CTC-ITB-01.**

A   ICC staining of CTC-ITB-01 for ß-catenin (green), E-cadherin (orange), and DAPI (blue). The scale bar corresponds to 20 μm. Two representative panels are shown for CTC-ITB-01.
B   ICC staining of CTC-ITB-01 for α-catenin (green), visualization of actin filaments by phalloidin (orange), and nuclei by DAPI (blue). The scale bar corresponds to 20 μm. Two representative panels are shown for CTC-ITB-01.
C   Generic EMT score of the adherent and non-adherently growing CTC-ITB-01 fractions in relation to tumor samples from different cancer entities. Scaling ranges from 1 (completely mesenchymal) to −1 (entirely epithelial). The adherent fraction is shown with a black dot, the non-adherent fraction with a black triangle, and the mean score per cancer entity by a black line.
D   Generic EMT score of the adherent and non-adherently growing CTC-ITB-01 fractions in relation to established cancer cell lines. Scaling ranges from 1 (completely mesenchymal) to −1 (entirely epithelial). The adherent fraction is shown with a black dot, the non-adherent fraction with a black triangle, and the mean score per cancer entity by a black line.

Source data are available online for this figure.

many CTCs with an epithelial phenotype. However, we could not exclude that mesenchymal CTC phenotypes lacking EpCAM or keratin expression were also present in the blood of this patient but remained undetected by CellSearch®, as shown in previous CTC studies (de Wit *et al*, 2018; Keller *et al*, 2019). To avoid selection bias for a particular phenotype of CTCs (Alix-Pana-bieres & Pantel, 2014a), we therefore took another blood sample from the same patient and cultured CTCs that were enriched by depletion of leukocytes using the Rosette Sep technology allowing an enrichment independent from the CTC phenotype. Thus, CTCs with epithelial and mesenchymal attributes had the same chance to be enriched and grown in culture.

This might explain why the morphology of the CTC-ITB-01 cell line is heterogeneous and not typical for an epithelial-like breast cancer cell line such as MCF-7. CTC-ITB-01 cells grow in parallel as adherent and non-adherent cell fractions (Fig 1C and D) and in varying cell sizes (Appendix Fig S3). The non-adherent cells grow out from the adherent cells into the medium resembling a string of pearls. Cultivating of either fraction separately results in adherently growing cells and subsequently similarly gives rise to the development of cells in suspension, indicating a high plasticity of CTC-ITB-01 cells (Pei *et al*, 2019).

The epithelial–mesenchymal morphology is characterized by a mixed population of epithelial cells in close cell–cell contact together with elongated, spindle-like cells growing disconnectedly (Fig 5A and B).

The expression of epithelial cell adhesion proteins detected via immunocytochemistry in adherently growing CTC-ITB-01 cells mirrors this phenotypic heterogeneity. While a subpopulation of these cells displays strong membranous E-cadherin staining with beta-catenin co-localization (Fig 5A), others show a more diffuse and weak expression of these proteins. Alpha-catenin displays

similarly diverse expression patterns (Fig 5B). Interestingly, E-cadherin appears to aggregate in small filaments, visible in all cells (Fig 5A).

To provide evidence for EMT features of our cell line, we analyzed the expression of the EMT transcription factors TWIST, SLUG, and SNAIL by Western blot analysis. Compared to MCF-7, increased levels of TWIST and SLUG were observed, while SNAIL was rather weaker expressed in CTC-ITB-01 (Fig 4A), providing further evidence that CTC-ITB-01 cells exhibit some signs of EMT.

To gain deeper insights into the EMT status of both cellular fractions, CTC-ITB-01 cells were assessed by analyzing RNA-sequencing data (from biological triplicates) using an EMT-scoring algorithm (Tan *et al*, 2014). This classifier compares data gained from hundreds of cancer cell line and tumor samples to allocate a score between -1 (completely epithelial) and +1 (completely mesenchymal) on the EMT spectrum. Both the adherent ($\bar{x} = -0.619$, s = 0.002) and non-adherent ($\bar{x} = -0.661$, s = 0.011) CTC fractions were clearly designated epithelial (Fig 5C and D). Compared with clinical breast cancer tumor tissue samples (Fig 5C) as well as other cancer cell lines (Fig 5D), CTC-ITB-01 appears to fall on the more epithelial end of the EMT spectrum, well below the established median value in both cases.

## Xenograft formation and metastasis in immunocompromised mice

To assess tumorigenic as well as metastatic potential of the CTC-ITB-01 line, a GFP-luciferase transduced cell population was directly injected into the mouse milk duct system of four 10-week-old immunodeficient NOD scid gamma (NSG) females. This has proven to be an efficient model for ER$^+$ cancer cell lines, allowing outgrowth under physiological systemic estrogen and low SLUG

**Figure 6. Xenograft of CTC-ITB-01 in immunodeficient mice confirms tumorigenic and metastatic potential.**

A   *In vivo* bioluminescence imaging of CTC-ITB-01 in four NSG mice. Blue indicates low radiance, red indicates high radiance.
B   Logarithmic plotting of *in vivo* radiance increase over time. Curves represent means ± SEM of measurements performed on multiple glands (*n* = 8).
C   H&E staining of xenografted mouse mammary glands show formation of primary tumors.
D   Stereoscope image of a mammary gland of immunodeficient NSG mouse. Fluorescently tagged CTC-ITB-01 cells in green.
E   IHC staining for ER, confirms retained *in vivo* ER positivity of formed tumors. Representative images of ER staining in higher (left panel) and lower (right panel) magnification of 2 different areas of a xenografted mammary gland.
F   Representative *ex vivo* luminescence images of indicated organs from engrafted mice. Organs include brain (1), lung (2), liver (3), and bones (4).
G   Scatter plot showing *ex vivo* radiance intensity of metastatic organs in three different mice (M1, M2, M3), indicating highest intensity in liver and bone. Data represent means ± SD of measurements performed on metastatic organs derived from three different mice.

Source data are available online for this figure.

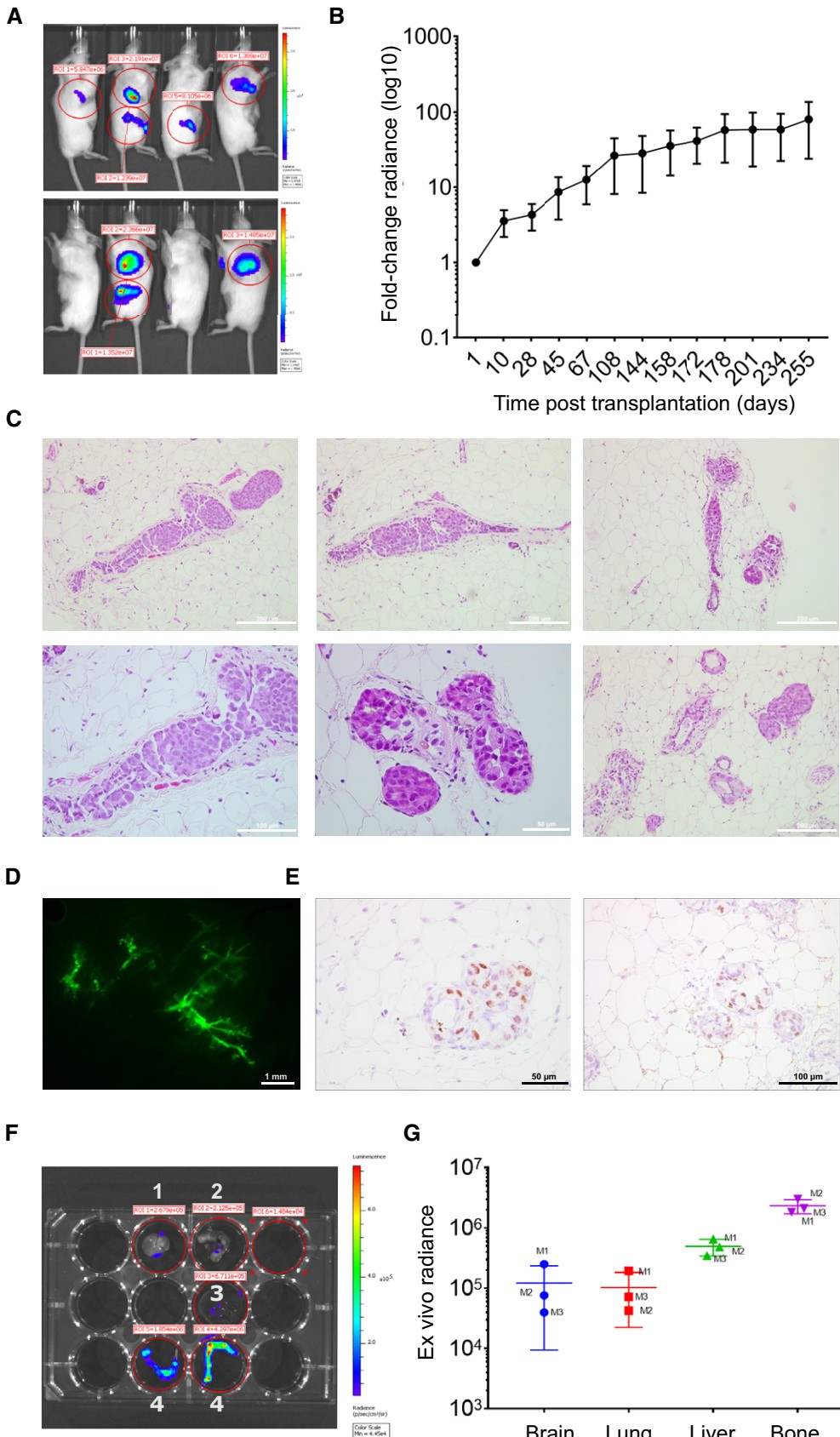

Figure 6.

levels (Sflomos *et al*, 2016; Ozdemir *et al*, 2018). Mice were monitored by *in vivo* bioluminescence imaging to assess the formation and proliferation of primary tumors and spontaneous metastases in clinical relevant tissues.

Primary tumor formation was seen in all mice bearing intraductally injected glands (Fig 6A). Tumor burden increased steadily over time until sacrificing after 8.5 months, assessed by the fold change radiance in bioluminescence imaging *in vivo* (Fig 6B). H&E sections of the mouse milk duct tissue (Fig 6C) confirmed the tumorigenic potential of CTC-ITB-01 in NSG mice. Fig 6D shows a stereoscope image of intraductal fluorescently tagged CTC-ITB-01 cells. Immunohistochemical staining for ER furthermore revealed that the ER$^+$ status of the cell line was maintained on the CTC-derived xenografts (CDX), confirming that histopathologic features are preserved (Fig 6E). Upon sacrificing, three out of four mice displayed micrometastases in the brain, lung, bone, and liver (Fig 6F and G). The fourth mouse had to be sacrificed early due to health reasons unrelated to tumor growth. The overall highest *ex vivo* radiance was detected within the bone metastasis of the mice, mirroring the predominant metastatic site of ER$^+$ breast cancer patients (Fig 6G). Interestingly, both bone and liver represented metastatic sites in the patient who harbored the CTCs that gave rise to CTC-ITB-01 and displayed high *ex vivo* bioluminescence in the MIND mouse model.

Conclusively, CTC-ITB-01 cells have tumorigenic potential in NSG mice upon intraductal injection and metastasize to clinically relevant organs such as bone, liver, brain, and lung.

## Estrogen sensitivity and relevance of the ER-alpha for growth of CTC-ITB-01

CTC-ITB-01 cells were isolated after the patient progressed under two different endocrine therapy regiments based on the aromatase inhibitor letrozole and later tamoxifen (Fig 1A). Therefore, CTC-ITB-01 cells were tested for their ability to respond to 17-β-estradiol (E2). To rule out unspecific effects of estrogens and other substances which could cause activation of ER-alpha within the culture medium, the experiment was carried out using phenol red-free RPMI substituted with charcoal-stripped FBS. When cultured under estrogen-free conditions or in the presence of E2 concentrations $\leq 10^{-11}$, CTC-ITB-01 cells showed slower outgrowth than when cultured in the presence of higher E2 concentrations (Fig 7A right panel). This indicates that the ER-alpha within CTC-ITB-01 cells is still functional and high estrogen concentrations cause accelerated cell growth. No differences in the morphology of CTC-ITB-01 cultured in the absence or in the presence of different E2 concentrations were observed (Fig 7A left panel). The fact that CTC-ITB-01 cells still grow and show no significant morphological changes under estrogen-free conditions point to these cells being able to tolerate estrogen deprivation and therefore endocrine regiments.

To investigate the potential relevance of the ER-alpha for outgrowth of CTC-ITB-01, stable knockdowns of ER-alpha with two different shRNAs were performed. Knockdown of ER-alpha with both shRNAs showed strong reduction of colonies in comparison with non-targeted shRNAs in colony-forming assays (Fig 7B left upper and lower panel). Efficiency of ER-alpha knockdowns were evidenced by Western blotting (Fig 7B right panel). This demonstrates that the ER-alpha is relevant for growth of CTC-ITB-01 under the given culture conditions. To investigate potential differences in the functionality of ER-alpha between CTC-ITB-01 and MCF-7 cells, both cell lines were cultured under identical conditions within estrogen-free, charcoal-stripped, phenol red-free medium. Both cell lines were cultured under E2-free conditions or incubated in the presence of E2 as indicated. After 24 h cells were harvested and differences in expression levels of certain ER-alpha target genes were tested by Western blotting. MCF-7 cells express ER-alpha target genes only in the presence of E2, whereas CTC-ITB-01 cells continue to express these genes in the absence of E2 (Fig 7C). This data suggests that ER-alpha within CTC-ITB-01 cells is active even in the absence of E2, which might explain why CTC-ITB-01 can tolerate low E2 levels. Interestingly, while MCF-7 expresses PR-A (progesterone receptor α) and B (progesterone receptor β) in an E2-dependent manner, CTC-ITB-01 cells display low amounts of PR-A but not B. However, both cell lines show reduced levels of ER-alpha in the presence of E2 indicating that both cell lines have functional mechanisms that regulate ER-alpha expression under the influence of estrogen (Martin *et al*, 1993).

**Figure 7. Estrogen and Palbociclib sensitivity of CTC-ITB-01.**

A Equal numbers of CTC-ITB-01 cells were seeded on 6-well culture plates and grown in the presence or absence of different 17-β-estradiol concentrations as indicated. After 10 days, cells were fixed and stained with Coomassie blue staining (right panel). Macroscopic photographs are shown of the fixed cell colonies from cells grown in the absence or presence of the indicated E2 concentrations (right panel). Scale bar 200 μm.

B Stable knockdown of ER-alpha within CTC-ITB-01 cells was performed by using lentiviral transfer of non-targeted (scramble) or two different targeted shRNAs against ER-alpha. CTC-ITB-01 cells were seeded on 6-well plates. After 10 days, cells were fixed and stained (left upper panel). Macroscopic photos are shown of the fixed cells (left lower panel), scale bar 200 μm. Western blots of lysates from knockdown of ER-alpha expression in CTC-ITB-01 cells transduced with the same amounts of lentiviral vectors were probed with the indicated antibodies. The extracts were obtained 72 h after lentiviral transduction (right panel). Where indicated, Western blots were visualized at long exposure (l.e.) and/or short exposure (s.e.).

C CTC-ITB-01 and MCF-7 cells were grown under the same culture conditions in the presence or absence of E2 as indicated. 24 h after E2 deprivation or E2-treatment cells were harvested. Western blots of the protein lysates were probed with antibodies targeting ER-alpha, FOXM1, Bcl-2, PR-A, PR-B and ID1, actin was used as loading control, visualized at long exposure (l.e.) and/or short exposure (s.e.).

D, E (D) Growth curves of CTC-ITB-01 and (E) MCF-7 cells under varying concentrations of Palbociclib treatment were measured using an IncuCyte Zoom live cell imaging system. The mean values from three technical triplicates (one experiment) with standard deviation are shown.

F, G (F) Influence of Palbociclib on CTC-ITB-01 growth after 300 h and (G) MCF-7 after 80 h. Data were chosen from one representative time point during the exponential growth phase. Statistical significance was analyzed with a one-way ANOVA with Dunnett's multiple comparisons test and compared to the vehicle substance dimethyl sulfoxide (DMSO). The mean values from technical triplicates (one experiment) with standard deviation are shown.

H, I (H) Effect of Palbociclib on the growth of CTC-ITB-01 cells and (I) MCF-7 cells. Concentrations were transformed to common logarithm. Three-parameter non-linear logistic regression was used to determine the IC$_{50}$. The mean values from three technical triplicates (one experiment) with standard deviation are shown. Error bars for standard deviation smaller than the symbols are not displayed.

Source data are available online for this figure.

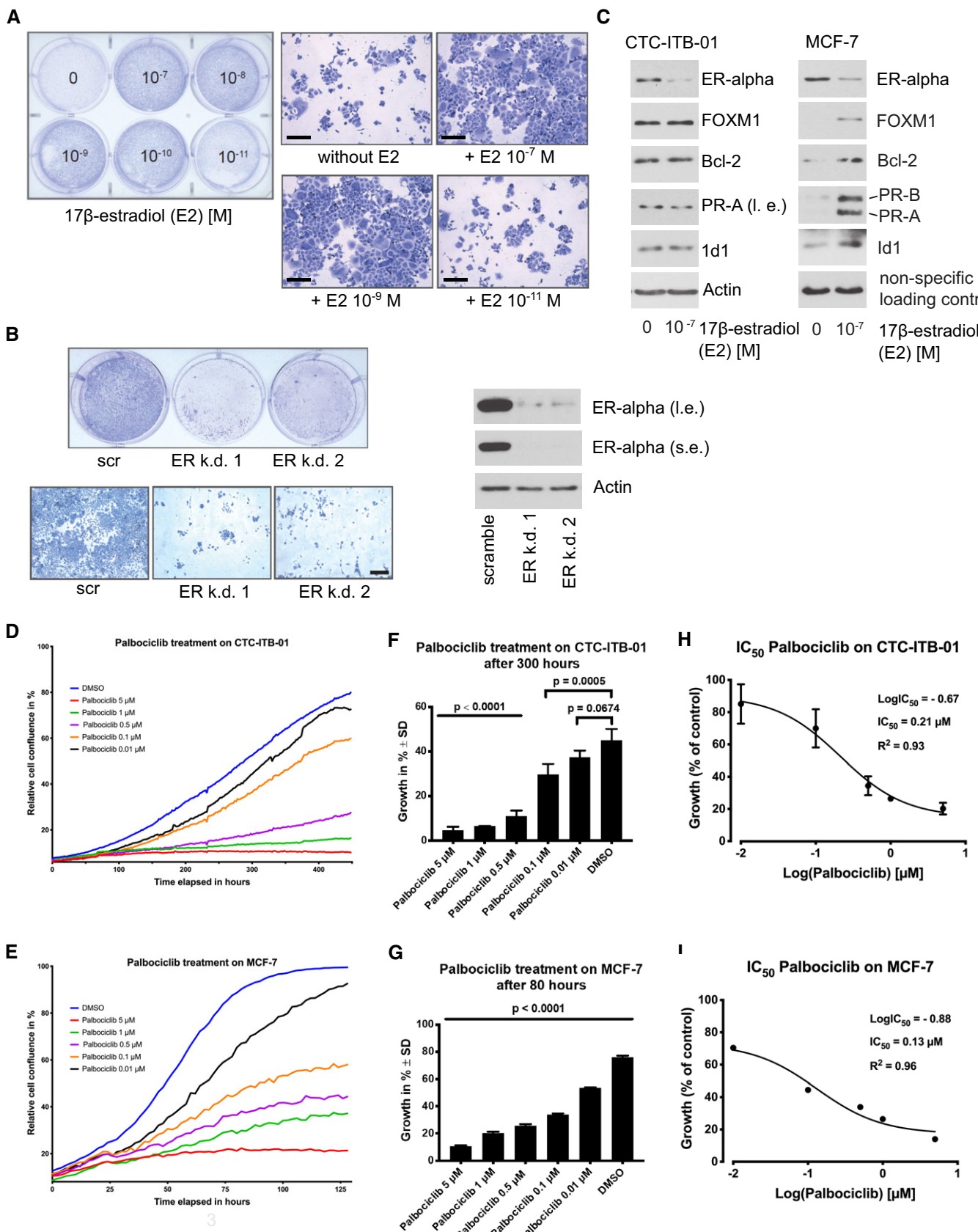

Figure 7.

## Response of CTC-ITB-01 to cyclin-dependent kinase (CDK) 4/6 inhibitors

While the patient with breast cancer giving rise to CTC-ITB-01 progressed on endocrine and later chemotherapy, the added benefit of novel cyclin-dependent kinase (CDK) 4/6 inhibitors has only recently been described for ER$^+$ breast cancer patients (Finn et al, 2015, 2016). At the time point when the patient had been treated, CDK inhibitors were not available for standard of care therapy in patients with breast cancer in Germany. To test the potential sensitivity of CTC-ITB-01 toward CDK4/6-inhibitors, CTC-ITB-01 and MCF-7 cells were treated with different concentrations of Palbociclib (Fig 7D and E). Cell growth is significantly reduced in CTC-ITB-01 already at a dose of 0.1 μM of this CDK4/6 inhibitor ($P < 0.001$) in relation to the DMSO control (Fig 7F). Compared to the MCF-7 cell line, CTC-ITB-01 shows similar sensitivity toward Palbociclib at lower inhibitor doses (e.g., 0.1 μM Palbociclib) and slightly increased sensitivity at higher doses of 0.5–5 μM (Fig 7F and G). The IC50 values were 0.21 μM for CTC-ITB-01 (Fig 7H) and 0.13 μM for MCF-7 (Fig 7I), respectively, indicating that the patient might have benefited from treatment with CDK4/6 inhibitors.

# Discussion

The functional properties of CTCs are still largely unknown due to the rarity of functional models such as cell lines derived from CTCs that mirror the in situ CTCs in patients with cancer. A frequent argument raised against the use of cell lines is that tumor cells may change their original characteristics rapidly in cell culture. Here we provide, for the first time, a direct genomic comparison of the original CTCs isolated from a patient with breast cancer to the cell line established from these CTCs. Our results demonstrate that our ER$^+$ breast CTC line resembles largely the CNA patterns of in situ CTCs, suggesting that the cell line has not undergone substantial genomic changes during the culturing period (passage 4 and passage 30) and might represent a good model for studying CTC biology.

The high number of CTCs isolated was certainly one precondition for the establishment of our CTC line. However, high CTC numbers on its own are no guarantee to obtain a permanent cell line (Tayoun et al, 2019; Faugeroux et al, 2020). Thus, we assume that the cells of our cell line have also the right biological properties of the most "aggressive" subset of CTCs. Future studies comparing CTC lines with CTCs that will stop proliferating after short term culture might help to define the molecular make up required for permanent survival and growth of CTCs.

CTC-ITB-01 cells showed a luminal B breast cancer subtype (ER$^+$, ERBB2$^-$); in particular the stable ER expression is remarkable in view of the paucity of ER$^+$ breast cancer cell lines. Thus, our CTC cell line provides a novel model to study ER$^+$ breast cancer CTCs. Moreover, for the first time, genomic aberrations of a breast cancer CTC-derived cell line were compared between the initially isolated CTCs, primary tumors and a distant metastasis.

The half-life of CTCs in the circulation is very short (few hours) (Meng et al, 2004) and the blood sample in our study was obtained years after primary tumor diagnosis in a highly metastatic setting. Thus, the CTC line is most likely derived from CTCs released from one or multiple metastatic sites (bone, lymph node, liver, vagina, or spleen) of our patient. While CTC-ITB-01 cells showed a CDH1 variant [c.1204G>A; p.D402N] in combination with loss of 16q (Ciriello et al, 2015), this variant differs from mutations of the left, primarily lobular tumor and the distant metastasis. Additionally, CTC-ITB-01 retains strong CDH1 expression, arguing against a lobular origin. However, CDH1 molecules expressed by the CTC-ITB-01 cell line are potentially adhesion-deficient (mutation in EC3), which might be the cause for the observation that subcellular distribution is disturbed. Furthermore, the CTC-ITB-01 cell line shares a mutation in the NF1 gene, which belongs to the 9 frequently mutated driver genes in hormone receptor-positive breast cancer (Bertucci et al, 2019), with the primary lobular tumor and the vaginal metastasis. Besides shared variations with the primary tumors, CTC-ITB-01 and the vaginal metastasis exhibited an additional, less frequent PIK3CA mutation (c.1252G>A; p.E418K, Table 1), located in the region encoding the C2 calcium/lipid-binding domain (Saal et al, 2005; Stemke-Hale et al, 2008). Taken together, we can speculate that the vaginal metastasis might have contributed to the pool of CTCs but it appears not be the exclusive source.

In addition, CTC-ITB-01 cells have accumulated large genomic areas showing increased LOH as compared to the primary tumors, including large parts of chromosome 17p (13.905–18.391.123) harboring the TP53 gene locus. In combination with the TP53 mutation of CTC-ITB-01, this provides hints for a role of TP53 in dissemination and survival of CTCs in the bloodstream.

The in-depth characterization of CTC-ITB-01 cells on RNA and protein levels demonstrated a predominantly epithelial phenotype; additionally, CTC-ITB-01 cells showed phenotypic and functional in vitro and in vivo characteristics of cancer stem cells. EMT is assumed to play a crucial role in migration of tumor cells (Alix-Panabieres et al, 2017) and has been implicated in cancer progression, metastasis (Chaffer et al, 2016) and resistance to chemotherapy (Nieto et al, 2016). It has furthermore been established that CTCs display plasticity on the EMT spectrum (Armstrong et al, 2011; Nieto et al, 2016; Alix-Panabieres et al, 2017), shifting between epithelial and mesenchymal states depending on therapy and stages of disease progression (Yu et al, 2013). However, EMT enables invasiveness while simultaneously reducing the self-renewal capacity of tumor cells, thus reducing their potential to outgrow at distant sites (Celia-Terrassa et al, 2012). Interestingly, CTC-ITB-01 and the in situ CTCs of our index patient expressed epithelium-specific proteins such as EpCAM and keratins. EMT-scoring, using an established algorithm further revealed that CTC-ITB-01 resides on the epithelial end of the EMT spectrum.

Interestingly, CTC-ITB-01 cells showed an interchangeable adherent and non-adherent cell population similar to the behavior described for cancer stem cells (Cariati et al, 2008). This growth pattern was consistent with the observed dysregulated subcellular distribution of cell adhesion proteins such as E-cadherin, alpha-, and beta-catenin, with a significant expression of some transcription factors (TWIST1, SLUG) known to be involved in EMT regulation as well as with activation of the NUMB pathway. Nevertheless, RNA sequencing of both adherent and non-adherent cell layer of CTC-ITB-01 showed predominantly an epithelial signature, which is consistent with a recent report (Ebright et al, 2020), indicating that proliferating CTCs with an epithelial signature were the most aggressive subset associated with an unfavorable clinical outcome in breast cancer.

There have been multiple studies linking EMT and stemness in CTCs, revealing that both cellular programs are closely intertwined (Raimondi et al, 2011; Papadaki et al, 2014). Originally a CD44[+]/CD24[−] phenotype was associated with breast cancer stemness (Al-Hajj et al, 2003). While the CTC-ITB-01 cell line was CD44[−]/CD24[+], it showed a high expression of ALDH1, a more recent breast cancer stemness marker (Ginestier et al, 2007; Charafe-Jauffret et al, 2010). It has been reported that CD44[−]/CD24[+] and ALDH1[+] tumor cells represent two distinct tumor cell populations not only localized in different regions of the primary tumor tissue but also expressing distinct EMT and MET gene profiles (Liu et al, 2014; Colacino et al, 2018). Furthermore, ALDH1-expressing stem cells have been described as epithelial and proliferative in contrast to the more mesenchymal and quiescent CD44[+]/CD24[−] stem cells (Ginestier et al, 2007; Liu et al, 2014; Colacino et al, 2018), which is consistent with the CD44[−]/CD24[+]/ALDH1[+] epithelial phenotype of CTC-ITB-01 cells that were capable of ex vivo growth. Although CTC-ITB-01 shows some signs of cancer stem cells, more detailed future investigations are required to further determine whether our CTC line fulfills all criteria of cancer stem cells.

Apart from intrinsic attributes of CTCs, the interaction with the surrounding microenvironment is crucial for successful extravasation and colonization of distant organs. The cancer secretome can therefore function as an indicator of how the CTCs attempt to manipulate the host tissue. Here, we analyzed selected secreted proteins, known to be involved in different aspects of tumor development, progression and metastasis via the EPISPOT assay (Ramirez et al, 2014; Cayrefourcq et al, 2015). Apart from K19, CTC-ITB-01 secreted VEGF, a known angiogenic inducer as well as promoter of cancer stem cell self-renewal (Beck et al, 2011; Saharinen et al, 2011; Zhao et al, 2015; Liu et al, 2017).

The tumorigenic nature of CTC-ITB-01 was further demonstrated in CDX models. Growing ER[+] cell lines as xenografts in vivo has not been successful without addition of exogenous 17β–estradiol in the past (Guiu et al, 2014; Sikora et al, 2014). This hormonal treatment, however, has detrimental effects on the health of the mice and induces estrogen levels equivalent to premenopausal conditions. It therefore does not adequately mirror disease development in the patient as most women develop ER[+] breast cancer in a post-menopausal setting. PDX are similarly difficult to establish displaying an engraftment rate of around 2.5% in immune-compromised mice (Cottu et al, 2012). This has contributed to underrepresentation of ER[+] tumors in mouse models due to a favoring of more aggressive histological subtypes (ERBB2[+] and triple-negative). Sflomos et al recently developed an elegant in vivo model circumventing many of these issues and allowing outgrowth of ER[+] cell lines under physiological hormone levels, thus increasing the engraftment rate to 30–100% (Sflomos et al, 2016). Using this model, we confirmed the tumorigenic capacity of CTC-ITB-01 in immunodeficient NSG mice. Apart from primary tumor formation in the mouse milk ducts of all four animals, three out of four mice displayed metastasis in organs equal to those of the patient with breast cancer, thus mirroring disease development in the patient with cancer. Additionally, all xenografts preserved the histopathological features (ER[+]) of their clinical counterpart.

As the CTC line was derived from a patient progressing first on hormone therapy and subsequently on chemotherapy, it could provide a valuable model to study resistance to therapy. While the ER remains partly functional and relevant for CTC-ITB-01 growth, downstream signaling appears to be constitutively active independent of ligand availability. This might provide a survival benefit for CTC-ITB-01 under extremely low ligand levels achieved by endocrine therapy, which might explain why the patient progressed under endocrine therapy. While ESR1 mutations represent a common mechanism of acquired resistance to endocrine therapy (Jeselsohn et al, 2015), we did not detect these mutations in CTC-ITB-01, suggesting a different mechanism of resistance in our index patient. Interestingly, treatment with the CDK4/6 inhibitor Palbociclib showed a concentration-dependent inhibition of CTC-ITB-01 growth, suggesting that the patient may have benefitted from this new line of therapy recently approved for ER[+] patients (Turner et al, 2015; Finn et al, 2016). However, our in vitro data should be further tested by in vivo xenotransplantation models.

In conclusion, we characterize a unique CTC-derived ER[+] breast cancer cell line that shares important features of in situ CTCs. This CTC line therefore represents a promising new tool for functional studies on CTC biology and response to novel drugs envisaged for ER+ breast cancer.

# Materials and Methods

### Blood collection and processing

Informed consent was obtained from all subjects, and the experiments conformed to the principles set out in the WMA Declaration of Helsinki and the Department of Health and Human Services Belmont Report as well as to the guidelines for experimentation with humans by the Chambers of Physicians of the State of Hamburg ("Hamburger Ärztekammer"). For this study, the blood of 50 patients with metastatic breast cancer was collected and processed. Samples were drawn into standard 7.5 ml ethylenediaminetetraacetic acid (EDTA) vacutainers or a CellSave® (Menarini Silicon Biosystems) preservation tube, respectively. The patients therefore provided a matched sample of blood in an EDTA (for in vivo culture) and CellSave® tube (CTC enumeration) for further analysis. The blood sample collected into the CellSave® preservation tube was processed with the CellSearch® system (Riethdorf et al, 2007). All analyses were performed by experienced scientists. CTCs were defined as keratin-positive, CD45-negative cells with a nuclear DAPI staining.

### Isolation and cultivation of CTC-ITB-01

CTCs were enriched from 7.5 ml of EDTA blood by Rosette Sep™ (StemCell Technologies) according to the manufacturer's instructions but using 20 μl of Rosette Sep™ solution per ml of blood instead of 50 μl/ml. The cell pellet gained from Rosette Sep™ enrichment was resuspended in 3 ml of RPMI complete medium. RPMI complete comprises RPMI 1640 (Gibco), 10% Fetal Calf Serum (FCS) (Gibco), 1% penicillin-streptomycin mix (Gibco), 1% L-glutamine (Gibco), 1% Insulin-Transferrin-Selenium-A Supplement (100X) liquid (Life Technologies), 10 ng/ml FGF2 (Miltenyi), 50 ng/ml EGF (Miltenyi), 0.1 μg/ml hydrocortisone (Sigma-Aldrich), and 0.2 μg/ml cholera Toxin (Sigma-Aldrich). Additional information is provided in Appendix Supplementary Methods. The cell solution

was distributed into a 96-well plate and cultured in standard cell culture conditions (37°C, 5% $CO_2$). After 14 days of cell culture, medium was changed. At 90% confluence, outgrowing cells were transferred to a 12-well cell culture dish and expanded.

## Standard cell culture

MCF-7 (ATCC® HTB-22), Hs578T (ATCC® HTB-126), SKBR3 (ATCC® HTB-30), MDA-MB-468 (ATCC® HTB-132), MDA-MB-231 (ATCC® HTB-26), NBTII (bladder tumor), and 11B (laryngeal squamous cell carcinoma) cell lines were maintained in DMEM (Dulbecco's Modified Eagle's medium, Gibco) with 10% of FCS and 1% penicillin/streptomycin. A549 cells were cultured in DMEM supplemented with 10 mM of HEPES, 10% of FBS, and 1% penicillin/streptomycin. CTC-MCC-41 cell line was sustained in micrometastatic medium (RPMI1640, Growth factors: EGF and FGF-2, Insulin-Transferrin-Selenium supplement, L-Glutamine).

## Karyotyping

For metaphase spreads, cells were treated with colcemid (0.02 μg/ml) overnight, incubated with 0.0075 M KCl, fixed with methanol/acetic acid (3:1), dropped onto wet slides, stained with 5% giemsa, and mounted with entellan before imaging with the Axioplan 2 microscope (Zeiss). 25 metaphases per experiment were counted.

## DNA isolation from cell culture and FFPE tissue

Genomic DNA (gDNA) was isolated from cell culture using the *Nucleoppin® Tissue Kit* (Macherey & Nagel) according to the manufacturer's instructions. gDNA from FFPE material was isolated with the *innuPREP DNA Micro Kit* (Analytik Jena) according to the manufacturer's instructions. To increase gDNA yield, the lysis step was extended overnight. DNA was stored at −20°C until further use.

## Next-generation sequencing and CNA profiling

Single or pools of cells were picked via micromanipulation using a fluorescence microscope and underwent whole genome amplification using the PicoPLEX DNA-seq kit followed by library preparation according to manufacturer's instructions (Takara Bio). Sequencing was performed using an Illumina NextSeq 550. Copy number aberrations of all CTCs were analyzed against 8 single leukocytes from the same patient (control cells) according to a previous publication (Heitzer et al, 2013).

## Whole-exome sequencing and WES data analysis

Exonic and adjacent intronic sequences were enriched from genomic DNA of the primary tumor, the vaginal metastasis and the CTC line using the Agilent SureSelect Human All Exon V6 enrichment kit and were run on an Illumina HiSeq4000 sequencer by the Cologne Center for Genomics (CCG). Data analysis and filtering of mapped target sequences were performed with the "Varbank" exome and genome analysis pipeline v.2.1 (CCG) and data were filtered for high-quality (coverage of more than 6 reads, a minimum quality score of 10) variants. WES data were analyzed for variants in 219 (Appendix Table S1) of the frequently mutated genes based on the COSMIC, HGMD and OMIM databases (Tate et al, 2019) and genes involved in hereditary cancer predispositions syndromes. We filtered for variants with an allele frequency ≥ 10% in at least one of the four samples, a minor allele frequency (MAF) < 1% in the gnomAD database (Karczewski et al, 2020), and a predicted mutational effect on the encoded protein. Silent mutations and intronic variants were included as long as a functional effect on splicing was predicted.

## Immunocytochemical staining

For ICC on cytospins and chamber slides, the following primary antibodies were applied: pan-keratin (Alexa488 or Alexa555 conjugated clone: AE1/AE3; eBioscience, 1:300), CD45 (Alexa647 conjugated, clone: Hi-30; BioLegend, 1:200), ER (clone SP1; Sigma-Aldrich, 1:100), ERBB2 (clone 29D8; Cell Signaling, 1:100), ALDH1 (clone 44; BD Pharmingen, 1:100), p53 (clone DO-1, Merck Millipore, 1:100) E-cadherin (Alexa555 conjugated, clone 36, BD Pharmingen, 1:200), α-catenin (clone D9R5E; Cell Signaling, 1:200), and ß-catenin (clone D10A8; Cell Signaling, 1:100).

CTC-ITB-01 cells as well as appropriate tumor cell line controls were spun down on glass slides (190 × g for 5 min). Having dried overnight, cells were fixed with 4% PFA (Sigma-Aldrich), permeabilized using 0.1% Triton (Sigma-Aldrich), and blocked with 10% AB-serum (Bio-Rad).

For chamber slide ICC staining, 30,000 CTC-ITB-01 cells were seeded into 2-well chamber slides (Nunc Lab-Tek Chamber Slide System, 2-well Permanox slide, Thermo Fisher Scientific) and incubated under standard culture conditions for 48 h. Subsequently, medium was discarded and cells were washed with 1× PBS (Gibco) three times prior to being fixed with 2% PFA (Sigma-Aldrich) for 10 min. Washing was repeated prior to permeabilization with 0.1% Triton X-100 (Sigma-Aldrich). Following another washing with 1× PBS, slides were blocked for 20 min using 10% AB-serum (Bio-Rad). Primary antibodies were incubated for 60 min.

Secondary antibodies for ICC on cytospins and chamber slides were applied for 60 min. Species-specific secondary antibodies consisted of rabbit-anti-mouse-A546 (polyclonal, Thermo Fisher Scientific), rabbit-anti-mouse-A488 (polyclonal, Thermo Fisher Scientific), and mouse-anti-rabbit-A488 (polyclonal, Thermo Fisher Scientific).

For multiple immunofluorescence stainings, either different fluorescently conjugated antibodies were used as cocktail or subsequently to the sequential application of an unconjugated primary antibody and a fluorescently labeled secondary antibody, additional primary antibodies directly conjugated with fluorochromes were added.

After a final washing step with 1× PBS, cytospins and chamber slides were covered in Prolong Gold Antifade Mountant (Thermo Fisher Scientific) and cover-slipped for analysis. For visualization of actin filaments, phalloidin (Alexa555 conjugated, Life Technologies, 1:100) was used. DAPI was used to counterstain nuclei. Immunofluorescence was evaluated with the fluorescence microscope (Axioplan 2, Zeiss).

Chromogenic immunocytochemical analysis was performed using the DAKO REAL™ EnVision Detection System Peroxidase/DAB+ Rabbit/Mouse kit (DakoCytomation). Cytospins were fixed with 2% PFA (Sigma-Aldrich) and permeabilized with 1% Triton for 10 min each. Following $H_2O_2$ blocking, cells were incubated with the P53 antibody diluted in 10% AB-serum (Bio-Rad) for 45 min prior to incubation with the peroxidase-labeled EnVision™

polymer coupled with goat anti-rabbit/mouse immunoglobulins (DakoCytomation) for 15 min at room temperature. Subsequently, cells were treated with DAB (3, 3 -diaminobenzidine) in substrate buffer containing hydrogen peroxide reagent for 10 min and cover-slipped using Eukitt mounting medium (Sigma-Aldrich). Immunostaining was evaluated using the microscope Axioplan 2 (Zeiss).

### Immunohistochemical assays

Patient as well as mouse-derived tumor tissues were formalin-fixed and paraffin-embedded (FFPE) for immunohistochemical analysis (IHC). Paraffin-embedded tissue specimens were taken from tissues acquired for routine diagnostic purposes at the Department of Pathology, University Medical Center Hamburg-Eppendorf. Immunohistochemical analysis was performed using the DAKO REAL™ Detection System Peroxidase/DAB+ Rabbit/Mouse kit (DakoCytomation). In brief, after pressure cooker pre-treatment of deparaffinized sections in citrate buffer (Bio-Genex Laboratories) for 5 min at 120°C and incubation with TBST (20 mM Tris, 150 mM NaCl, 0.1% Tween-20) for 5 min, the primary antibody (Mouse anti-E-cadherin, clone 36, BD Biosciences, final concentration: 1.25 µg/ml) was applied overnight at 4°C. Subsequent application of the biotinylated secondary antibodies for 10 min and endogenous peroxidase blocking solution for 5 min were followed by incubation with streptavidin peroxidase for 10 min. For visualization of E-cadherin-specific immunostaining, DAB diluted in substrate buffer containing hydrogen peroxide was utilized as chromogen. Counterstaining of nuclei was performed with Mayer's hemalum solution (Merck).

For mouse experiments, engrafted glands were dissected, fixed for 4 h in 4% paraformaldehyde, and paraffin-embedded. Four micrometer sections were cut, mounted onto 76 × 26 mm microscope slides (Rogo-Sampaic), and stained with hematoxylin–eosin (H&E). IHC was performed using estrogen receptor (cat#790-4324, clone SP1) by the Ventana-automated staining device (Ventana Medical Systems Inc, Roche AG).

### Western blot analysis

Cells were harvested and homogenized by ultrasonic treatment. Protein concentration was measured using the Pierce BCA Protein Assay Kit (Pierce). All samples were generated in biological triplicates. For Western blot analysis, 20 µg of total protein per sample (EpCAM: 40 µg of total protein per sample) was applied. The primary antibodies were utilized according to the instruction manual of the supplier using appropriate dilutions (Table EV4). The appropriate secondary antibodies conjugated with horseradish peroxidase (DakoCytomation) according to the species of the primary antibody were used at dilutions from 1:500 to 1:10,000 depending on the signal intensity. Protein bands were visualized using SignalFire™Plus ECL reagent (Cell Signaling Technology) and X-ray films (CEA) according to the instruction manual. More detailed information is given in Appendix Supplementary Methods.

### Erbb2 fish

*ERBB2* gene copy numbers of CTC-ITB-01 line cells were determined by fluorescence *in situ* hybridization using the *ERBB2/Cen17*

Zytovision probes (Zytomed) as described previously (Riethdorf *et al*, 2010). Briefly, after denaturation at 80°C for 2 min, dehydration in a series of ascending ethanol, and air-drying, cells were digested with a pepsin solution at 37°C for 7 min in a humidity chamber. Subsequently to renewed dehydration and air-drying, hybridization with the probe previously denatured for 7 min at 75°C was carried out at 37°C for 16 h. After different washing steps, dehydration, and air-drying, nuclei were stained with DAPI.

### ALDH activity measurement

Aldehyde dehydrogenase (ALDH) activity was measured using the ALDEFLUOR™ kit (StemCell™ Technologies) following the instruction of StemCell™ Technologies. Briefly, 5 µl of ALDEFLUOR™ Reagent was added to the cell suspension and half of the cell mixture was immediately transferred into another tube with ALDE-FLUOR™ DEAB Reagent (an inhibitor of ALDH activity) to serve as internal negative control. ALDH activity was analyzed in comparison with the internal negative control for each cell line. As recommended by the manufacturer, the A549 cell line was used as positive control.

### EPISPOT assay

The fluoro-EPISPOT assay is able to detect low concentrations of secreted/released/shed proteins from viable cells at the single cell level (Alix-Panabieres *et al*, 2009). Cells are cultured for a short time (48 h) on a membrane coated with antibodies against K19, VEGF, OPG, EGFR, or FGF2 aimed to capture the secreted/released/shed proteins. In the next step, a second either directly fluorescently conjugated antibody or biotin-labeled antibodies detecting the captured proteins are applied. Biotin-labeled antibodies are then detected by streptavidin conjugated to Alexa 555 (Invitrogen). For detailed information about the antibodies used, see Appendix Supplementary Methods. Single fluorescent immunospots were evaluated under an ELISPOT reader (C.T.L ImmunoSpot).

### Assessment of cell size

For immunofluorescence staining, cells were seeded on culture slides. Cells were fixed, permeabilized, and blocked overnight. Cytoskeleton was detected using anti-pan-keratin eFluor 570 clone AE1/AE3 (eBioscience, 1:80) and counterstained with DAPI. Fluorescence images were captured using a Zeiss Axioplan 2 epifluorescence microscope equipped with a charge-coupled device camera and AxioVision software. Cell size was measured using ImageJ.

### RNA Sequencing and bioinformatics analysis

RNA sequencing was performed at the NGS Integrative Genomics Core Unit (NIG), University Medical Center, Goettingen, in biological triplicates for all CTC samples. RNA was extracted using the NucleoSpin RNA Plus kit (Macherey Nagel). Following RNA extraction, quality and integrity of RNA were assessed with the Fragment Analyzer from Advanced Analytical by using the standard sensitivity RNA Analysis Kit (DNF-471). All samples selected for sequencing exhibited an RNA integrity number over 8 and RNA-seq libraries were generated using an mRNA-Seq protocol from Illumina, the

TruSeq stranded mRNA prep Kit (Cat. No. RS-122-2101), and quantified using the QuantiFluor™dsDNA System from Promega. The size of final cDNA libraries was determined and exhibited an average size of 300 bp. Libraries were pooled and sequenced on a HiSeq 4000 (Illumina) generating 50 bp single-end reads (30–40 Mio reads/sample).

Sequencing images obtained after sequencing in the form of.bcl files are demultiplexed and converted to fastq files using Illumina software bcl2fastq v2.17.1.14. Quality check of resulting sequence reads is performed using FastQC version 0.11.5. Generated raw reads are aligned to the reference Human genome version hg38 sourced from ENSEMBL database using aligner STAR version 2.5. BAM files generated from alignment step are used to perform read counting using software featureCounts from subread package version 1.5.0. Counts generated are used to perform the differential gene expression analysis in R environment version 3.4.3 loading R/Bioconductor package DESeq2 version 1.14.1. Candidate genes generated were filtered to a minimum log2fold change of 1 and a FDR-corrected $P$-value of 0.05. Heatmap/s depicting top 50 candidate genes are generated using R/Bioconductor package pheatmap version 1.0.8.

## Mouse model

CTC-ITB-01 cells were fluorescently tagged with a luciferase gene (GFP-fLuc2) using lentiviral technology. Efficiency of the lentiviral transduction lay at around 30–40%. A single-cell suspension of 300,000 cells was injected into the milk duct of NSG mice ($N = 4$). Mice were kept for 8.5 months prior to sacrificing. The tumor growth was monitored in real time at multiple time points following transplantation by Xenogen IVIS bioluminescence imaging system 200 (Caliper Life Sciences) in accordance with the manufacturer's recommendations and protocols 12–15 min after intraperitoneal administration of 150 mg/kg luciferin (cat# L-8220, Biosynth AG). Images were analyzed with Living Image software (Caliper Life Sciences, Inc.). For metastasis *ex vivo* detection, mice were injected with luciferin 10 min prior to sacrifice and resected organs were imaged immediately.

## Colony-forming assay

For colony-forming assays, cells were seeded at the same density per well of 6-well cell culture plates. Cells were fixed with 70% ethanol and stained with Coomassie blue staining solution.

## Viral transduction

The packaging cell line HEK293T was used for generation of lentiviruses (shRNA: NM_000125 TRCN0000003298 pLKO.1 CMV-tGFP, NM_000125 TRCN0000003300 pLKO.1 CMV-tGFP, Nontarget pLKO.1 CMV-tGFP) following standard calcium phosphate protocol. Transduction efficiency was monitored by flow cytometric detection of EGFP und efficiency of knockdown was tested by Western blotting.

## *In vitro* drug testing of Palbociclib

CTC-ITB-01 or MCF-7 cells were seeded in a density of $2 \times 10^3$ cells per well in technical triplicates into a flat bottom 96-well plate and

### The paper explained

#### Problem

Metastases developing upon dissemination of tumor cells from the primary tumor, access to the blood circulation and outgrowth at distant organs are the most frequent causes of breast cancer-related death. The vast majority of breast cancer cases is driven by hormone receptors enabling endocrine treatment as promising therapeutic strategy. However, tumor cells can develop resistance to this therapy, which constitutes a significant clinical problem, especially in the metastatic stage of the disease. Analyzing circulating tumor cells (CTCs) might be helpful to identify new therapeutic targets, but functional studies giving insight into the biology of CTCs are limited due to their low frequency and the lack of appropriate models.

#### Results

This study describes the establishment and characterization of a novel CTC-derived estrogen receptor (ER)-positive breast cancer cell line from a patient with metastatic ER-positive breast cancer, designated CTC-ITB-01. Downstream ER signaling is constitutively active in CTC-ITB-01 independent of ligand availability. This ER-positive cell line is resistant to endocrine therapy; however, the CDK4/6 inhibitor Palbociclib strongly inhibits CTC-ITB-01 growth. Genomic analyses revealed high concordance between CTC-ITB-01 and CTCs originally present in the patient with cancer at the time point of blood draw. Primary tumor and metastasis formation in an intraductal PDX mouse model reflected the clinical progression of ER-positive breast cancer.

#### Impact

Our work established and characterized a novel CTC cell line that mirrored *in situ* CTCs. This cell line allows first in-depth insights into the functional properties of CTCs in the most common ER+ breast cancer subtype and enables further experimental steps to uncover resistance mechanisms and to identify new therapeutic targets.

placed in the incubator at 37°C with 5% $CO_2$ overnight to adhere. The next day, Palbociclib (PD-0332991, Cayman Chemical), dissolved in DMSO, was serially diluted to the desired concentration in standard cell culture medium and added to the cells. Afterward, the plate was immediately transferred into the IncuCyte Zoom live cell imaging system (Essen Bioscience) and relative cell confluence was measured every 2 h. The processing definition and confluence mask of the IncuCyte system were created using the provided IncuCyte Zoom 2016B software (Essen Bioscience). Accuracy of fit of the confluence mask was verified on representative images at different states of growth during the experiment.

Statistical testing was performed with Prism 7.0a (GraphPad Software Inc.) using a one-way ANOVA followed by Dunnett's multiple comparisons test. A $P < 0.05$ was considered as statistically significant. $IC_{50}$ calculation was also achieved with Prism 7.0a using the implemented non-linear three-parameter logistic curve regression. Concentrations were transformed to common logarithm.

## Data availability

NGS data are deposited at the European Nucleotide Archive with the following accession number: PRJEB37968 (http://www.ebi.ac.uk/ena/browser/view/PRJEB37968).

**Expanded View** for this article is available online.

## Acknowledgements

The authors acknowledge Malgorzata Stoupiec, Cornelia Coith, Antje Andreas, and Olivier Mauermann from UKE Hamburg, and Alexandra Soler from CHU Montpellier for technical assistance. For preparation of the graphical synopsis, the figures were designed with the assistance of and modification to the images provided by the Servier Medical Art database http://smart.servier.com/. Further information pertaining to the license and disclaimer notices can be found here: https://creativecommons.org/licenses/by/3.0/. Klaus Pantel and Catherine Alix-Panabières received funding from the European IMI research project CANCER-ID (115749-CANCER-ID). Klaus Pantel and Sabine Riethdorf received funding from the Deutsche Krebshilfe (Nr. 70112504). Klaus Pantel has received research funding from the Deutsche Forschungsgemeinschaft (DFG) SPP2084 μBone and ERC Advanced Investigator Grant INJURMET (Nr. 834974). Cathrin Brisken received a grant from the Swiss Cancer Ligue KFS-3701-08-2015, Sonja Thaler obtained funding from the Deutsche Forschungsgemeinschaft (DFG grant TH 1523/3-1 to S Thaler) and George Sflomos was supported by the Biltema and ISREC Foundation.

## Author contributions

Conception and design: KP. Development and methodology: CK; AK; SR; KP. Acquisition of data: CK; AK; SAJ; SG; GY, GS; ST; DS; SG; PMM; LB; LC; JA; GS-R; KR; AZ; YG; KBa; LO; QZ; SR; SW. Analysis and interpretation of data: CK; AK; SAJ; GY; VM; GS; ST; DJS; KBa; LC; KBo; YG; TZT; MRS; LB; TMG; MJ; JPT; CB; SR; SW; CAP; KP. Administrative, technical, or material support: CK; SAJ; AK; MRS; VM; TMG; MJ; J-PT; CB; CA-P; KP. Study supervision: KP; SR; CK; AK; TMG. Writing, review, and/or revision of the manuscript: All authors. Final approval: All authors.

## Conflict of interest

KP and CAP have ongoing patent applications related to CTCs. KP has received honoraria from Agena, Novartis, Roche, and Sanofi and research funding from European Federation of Pharmaceutical Industries and Associations (EFPIA) partners (Angle, Menarini and Servier) of the CANCER-ID program of the European Union–EFPIA Innovative Medicines Initiative. CAP has received honoraria from Janssen and grant support from Menarini. The remaining authors declare no conflict of interest.

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
