## [Review Process File · EMBO Molecular Medicine]

Characterization of circulating breast cancer cells with tumorigenic and metastatic capacity

Claudia Koch, Andra Kuske, Simon Joosse, Goekhan Yigit, George Sflomos, Sonja Thaler, Daniel Smit, Stefan Werner, Kerstin Borgmann, Sebastian Gärtner, Parinaz Mossahebi-Mohammadi, Laura Battista, Laure Cayrefourcq, Janine Altmüller, Gabriela Salinas-Riester, Kaamini Raithatha, Arne Zibat, Yvonne Goy, Leonie Ott, Kai Bartkowiak, Tuan Zea Tan, Qing Zhou, Michael Speicher, Volkmar Müller, Tobias Gorges, Manfred Jücker, Jean Paul Thiery, Cathrin Brisken, Sabine Riethdorf, Catherine Alix-Panabieres, and Klaus Pantel

DOI: [10.15252/emmm.201911908](https://doi.org/10.15252/emmm.201911908)

Corresponding authors: Klaus Pantel (pantel@uke.de)

Review Timeline:

Submission Date:	16th Dec 19
Editorial Decision:	5th Feb 20
Revision Received:	5th May 20
Editorial Decision:	27th May 20
Revision Received:	15th Jun 20
Accepted:	17th Jun 20

Editor: Lise Roth

Transaction Report:

5th Feb 2020

Dear Dr. Pantel,

Thank you for the submission of your manuscript to EMBO Molecular Medicine, and please accept my apologies for the delay in getting back to you, which is due to the fact that it was difficult to secure referees over the holiday season. We have now received feedback from the three reviewers who agreed to evaluate your manuscript. As you will see from the reports below, the referees acknowledge the interest of the study and are overall supporting publication of your work pending appropriate revisions.

Addressing the reviewers' concerns in full will be necessary for further considering the manuscript in our journal, and acceptance of the manuscript will entail a second round of review. EMBO Molecular Medicine encourages a single round of revision only and therefore, acceptance or rejection of the manuscript will depend on the completeness of your responses included in the next, final version of the manuscript. For this reason, and to save you from any frustrations in the end, I would strongly advise against returning an incomplete revision.

When submitting your revised manuscript, please carefully review the instructions that follow below. Failure to include requested items will delay the evaluation of your revision:

- 1) A .docx formatted version of the manuscript text (including legends for main figures, EV figures and tables). Please make sure that the changes are highlighted to be clearly visible.
- 2) Individual production quality figure files as .eps, .tif, .jpg (one file per figure).
- 3) A .docx formatted letter INCLUDING the reviewers' reports and your detailed point-by-point responses to their comments. As part of the EMBO Press transparent editorial process, the point-by-point response is part of the Review Process File (RPF), which will be published alongside your paper.
- 4) A complete author checklist, which you can download from our author guidelines (<https://www.embopress.org/page/journal/17574684/authorguide#submissionofrevisions>). Please insert information in the checklist that is also reflected in the manuscript. The completed author checklist will also be part of the RPF.
- 5) Before submitting your revision, primary datasets produced in this study need to be deposited in an appropriate public database (see <https://www.embopress.org/page/journal/17574684/authorguide#dataavailability>). Please remember to provide a reviewer password if the datasets are not yet public. The accession numbers and database should be listed in a formal "Data Availability " section (placed after Materials & Method). Please note that the Data Availability Section is restricted to new primary data that are part of this study.

6) We would also encourage you to include the source data for figure panels that show essential data. Numerical data should be provided as individual .xls or .csv files (including a tab describing the data). For blots or microscopy, uncropped images should be submitted (using a zip archive if multiple images need to be supplied for one panel). Additional information on source data and instruction on how to label the files are available at .

7) Our journal encourages inclusion of *data citations in the reference list* to directly cite datasets that were re-used and obtained from public databases. Data citations in the article text are distinct from normal bibliographical citations and should directly link to the database records from which the data can be accessed. In the main text, data citations are formatted as follows: "Data ref: Smith et al, 2001" or "Data ref: NCBI Sequence Read Archive PRJNA342805, 2017". In the Reference list, data citations must be labeled with "[DATASET]". A data reference must provide the database name, accession number/identifiers and a resolvable link to the landing page from which the data can be accessed at the end of the reference. Further instructions are available at .

8) We replaced Supplementary Information with Expanded View (EV) Figures and Tables that are collapsible/expandable online. A maximum of 5 EV Figures can be typeset. EV Figures should be cited as 'Figure EV1, Figure EV2' etc... in the text and their respective legends should be included in the main text after the legends of regular figures.

- Additional Tables/Datasets should be labeled and referred to as Table EV1, Dataset EV1, etc. Legends have to be provided in a separate tab in case of .xls files. Alternatively, the legend can be supplied as a separate text file (README) and zipped together with the Table/Dataset file. See detailed instructions here: .

9) The paper explained: EMBO Molecular Medicine articles are accompanied by a summary of the articles to emphasize the major findings in the paper and their medical implications for the non-specialist reader. Please provide a draft summary of your article highlighting

10) For more information: There is space at the end of each article to list relevant web links for further consultation by our readers. Could you identify some relevant ones and provide such information as well? Some examples are patient associations, relevant databases, OMIM/proteins/genes links, author's websites, etc...

11) Every published paper now includes a 'Synopsis' to further enhance discoverability. Synopses

are displayed on the journal webpage and are freely accessible to all readers. They include a short stand first (maximum of 300 characters, including space) as well as 2-5 one-sentences bullet points that summarizes the paper. Please write the bullet points to summarize the key NEW findings. They should be designed to be complementary to the abstract - i.e. not repeat the same text. We encourage inclusion of key acronyms and quantitative information (maximum of 30 words / bullet point). Please use the passive voice. Please attach these in a separate file or send them by email, we will incorporate them accordingly.

Please also suggest a striking image or visual abstract to illustrate your article. If you do please provide a jpeg file 550 px-wide x 400-px high.

12) As part of the EMBO Publications transparent editorial process initiative (see our Editorial at <http://embomolmed.embopress.org/content/2/9/329>), EMBO Molecular Medicine will publish online a Review Process File (RPF) to accompany accepted manuscripts.

In the event of acceptance, this file will be published in conjunction with your paper and will include the anonymous referee reports, your point-by-point response and all pertinent correspondence relating to the manuscript. Let us know whether you agree with the publication of the RPF and as here, if you want to remove or not any figures from it prior to publication.

I look forward to receiving your revised manuscript.

Yours sincerely,

Lise Roth

Lise Roth, PhD
Editor
EMBO Molecular Medicine

To submit your manuscript, please follow this link:

Link Not Available

***** Reviewer's comments *****

Referee #1 (Remarks for Author):

The manuscript is of good technical quality, although, in reviewer's opinion the most complete and well performed part is the characterization of the established CTC cell line (CTC-ITB-01) in vivo. Conversely, the characterization of CTC-ITB-01 cells in vitro in order to establish the grade of tumor heterogeneity in culture and the stemness is lacking in some aspects.

It is not novel (see previous papers i.e. Aceto et al. Cell. 2014 Aug 28;158(5):1110-1122) but it provides an additional contribution to address the issue of Circulating Tumor Cells (CTCs) which is still largely unknown. In particular, it provides a novel model to study ER+ breast cancer CTCs Thus it may be of impact for further development of novel anti-cancer therapies.

Therefore, Reviewer believes that the manuscript is suitable for publication after a minor revision. Following a detailed analysis of the manuscript highlighting the main concerns that should be addressed.

In this manuscript authors provide an in-deep characterization of CTC-ITB-01 cells, a cell line established by isolation of CTCs from whole blood of a metastatic breast cancer patient. They show that CTC-ITB-01 cells show an epithelial ER+ positive phenotype and suggest that this cell line may reveal a novel suitable model to study ER+ breast cancer CTCs.

In this manuscript authors evaluate:

- morphology of cells
- copy number alterations (CAN),
- expression-based subtyping using the PAM50 and scmod2 classifiers.
- evaluation of Stem markers frequently showed by cancer cells endowed with high phenotypic plasticity. (CD44, CD24, ALDH, E-cadherin, alpha-catenin, EMT-scoring algorithm analyzing RNA-sequencing data).
- in vivo functional characterization of tumorigenic and metastatic potential.
- evaluation of endocrine sensitivity and chemoresistance.

1. The first and main concern regards the method used for isolation of CTCs.

CTCs identification and count through CellSearch® technology is based on staining of an epithelial marker (EpCAM) allowing to distinguish CTCs as EpCAM+ cells from leukocytes which are EpCAM-. The signature based on EpCAM, pancytokeratins and CD45 makes the CellSearch® System a reliable instrument with prognostic value in metastatic cancer patients (breast, colon and prostate). However, as the authors say at page 10: "Since cell culture is not possible on the fixed CTC fraction from the CellSearch® cartridge, we took an additional, parallel blood sample from the same patient and cultured CTCs enriched by negative depletion of leukocytes which is unbiased for a particular phenotype of CTCs".

Reviewer believes it should be performed a cytofluorimetric analysis establishing the amount of EpCAM+, CD45+ and CD34+ cells upon the establishment of the CTC-ITB-01 cell line. This analysis could highlight if some endothelial progenitors may persist during culture of CTC-ITB-01 cells. This is also of particular importance in order to establish if VEGF identified by functional fluoro-EPISPOT assay (Supplementary Figure 6) is secreted by CTC-ITB-01 cells or, rather it is produced by endothelial progenitors.

2. The characterization of tumor heterogeneity due to expression of stem markers and EMT in CTC-ITB-01 cell line is not properly addressed. Authors stated that the "morphology of the CTC-ITB-01 cell line is heterogeneous and not typical for an epithelial-like breast cancer cell line such as MCF-7" (at the end of Page 10). However, Figure 1C show well-shaped epithelial cells and this heterogeneity does not appear clearly.

In addition, at the end of page 10 authors say that cultivating both adherent and non-adherent

cells separately "gives rise to the respective counterpart, indicating a high plasticity of CTC-ITB-01 cells similar to that recently reported for cancer stem cells [41]." However, in reviewer's opinion, all the stemness markers evaluated strongly suggest that CTC-ITB-01 cells have a well-defined epithelial phenotype: cortical E-cadherin and cortical beta-catenin localization, low expression of CD44, vimentin and N-cadherin, high expression of CD24, CK8-18 positivity associated to ER expression. Also the PAM-50 classifier shows a great alignment of CTC-ITB-01 with a luminal B subtype.

Thus heterogeneity based on morphology and epithelial-mesenchymal markers in CTC-ITB-01 seems to be lacking.

However, if authors want to evaluate stem-like properties of isolated CTC-ITB-01 cells they could try to obtain tumors after xenografts at limiting dilutions or evaluate activation of some stem pathways such as Notch, Numb, Hedgehog, excluding wnt signaling since cortical beta-catenin localization (Figure 5) suggests that it might be uninvolved.

In addition, to establish if an Epithelial Mesenchymal Transition Programme (EMT) is activated in CTC-ITB-01 cells, authors could evaluate expression of some EMT marker as Twist, Snail1, Slug, c-Met. Even if this markers could be expressed in CTC-ITB-01 cells just because they are metastatic cells with high migratory activity however this information could be useful for those researchers interested in the use of this cell line model.

Minor revision

1- In supplementary Figure 1 E-cadherin staining in Formalin-fixed paraffin-embedded (FFPE) tissue of both primary tumors was shown. Authors show exemplary IHC images of both the primary lobular and ductal tumors. They should provide a magnification of both picture allowing the identification of E-Cadherin localization in breast tissues, in addition they should provide also the staining in the normal breast tissues.

2- If the existence of a mixed population showing mesenchymal or epithelial morphology is proven then Figure 2C should be replaced because the cells in the picture clearly appear as epithelial cells.

3- The following paragraph at page 10 is not clear for the non-specialists, in reviewer's opinion it may be misleading: "The high CTC count of the CellSearch® analysis (1,547 CTCs per ml of blood) indicated that a large number of the in situ CTCs originally present in the blood of the cancer patient expressed epithelial markers (EpCAM and keratins) [40]. Since cell culture is not possible on the fixed CTC fraction from the CellSearch® cartridge, we took an additional, parallel blood sample from the same patient and cultured CTCs enriched by negative depletion of leukocytes which is unbiased for a particular phenotype of CTCs [6]. Thus, CTCs with some degree of phenotypic plasticity would have also been selected and had a chance to grow in culture.

4- In the caption of Supplementary Figure 5 there is a mistake in point (2) ALH1 rather the ALDH1

5- In the caption of Supplementary Figure 6 the meaning of T- e T+ is not described .

Referee #2 (Remarks for Author):

This manuscript reports the characterization of a new cell line obtained from circulating tumor cells of a breast cancer patient. It will constitute a very important tool to study tumor dissemination in luminal breast cancer. These results are quite remarkable as it is very difficult and rare to obtain cell line and PDX from estrogen positive breast cancers, even more from CTC and not tumor samples.

The characterization of the cell line is very comprehensive, from whole genome sequencing and extensive immune cytology description to PDX formation.

The results are well discussed by experts in the field.

Comments:

As the probability of success to obtain a cell line from CTC is very low, it would be of interest to know how many patients were screened before the success with this patient?

In the clinical presentation of the patient case, the percentage of estrogen and progesterone receptor positivity would be of interest to be provided in the text and not only in supplementary table.

P19, it is reported that genomic DNA from primary tumor and from the vaginal metastasis was used for genomic sequencing. However, there is no result reported of the vaginal metastasis in the genomic section of the results page 7. Was the vaginal metastasis biopsied and analyzed and was it compared to CTC and primary tumors profiles?

It would be interesting to give the date of treatment of the patient, in order to understand which treatment were not available like CDKi in this time.

In the results, Table 1 about LOH, could be transferred into the supplementary tables and a new table with a summary of the main clinical significant mutations detected by WGS extracted from the supplementary table would be of more interest.

In figure 2, CNA evaluation was performed on cell line culture, on primary tumor but also on CTC collected at the first blood draw. How were CTC counted to perform NGS on 1 or 5 or 10 CTC? Was it micropipetting or a specific device to perform NGS on isolated cells?

Mutation of CDH1 was reported. Is there any result to hypothesize that this mutation is subclonal in order to explain the expression of E-cadherin by the tumor?

Page 13, could the authors clarify what is PR A and B.

The authors have tested the impact of palbociclib, a CDK inhibitor, on the CTC-ITB1 cell line, which show some characteristics associated with resistance to endocrine treatment. The CDKi are particularly of interest if combined with endocrine treatment. This has not been tested. Is there a specific reason for this strategy of monotherapy and not a combination with tamoxifen or fulvestrant?

In the discussion or introduction about CTC clusters, the authors could quote recent work by Gkoutela S et al in Cell 2019.

In the discussion, there is no mention of ESR1 mutation as a frequent mechanism of resistance to anti hormonal treatment, more frequently when exposed to aromatase inhibitor.

Referee #3 (Comments on Novelty/Model System for Author):

Low novelty: breast CTC culture and xenografts have been achieved in the past by several other groups (e.g. Yu et al, Science 2014).

Referee #3 (Remarks for Author):

Koch, Kuske and colleagues describe the establishment and characterization of a novel CTC breast cancer cell line. They demonstrate that this ex vivo model retains ER positivity in culture and genomic (CNV) concordance with the original CTCs from the very same patient. They also suggest that the established CTC line presents with partial EMT and cancer stem cell features, and demonstrate the ability of these cells to form metastasis in a PDX model. Together, this manuscript provides an interesting new model to study CTC biology, but some considerations should be made:

- (1) Breast CTC culture and xenografts have been shown in the past, including ER+ models with metastatic ability in animal models (e.g., Yu Science 2014 among others). From the current submission it is unclear what is the major novelty of this paper, compared to previous studies.
- (2) The authors interpret the CNV results (Figure 2) as "tumor evolution during the course of the disease", "the CTC line derived from a subpopulation of CTCs", and "CTC line stability" depending on specific comparisons. However, they should exclude potential technical biases of this type of analysis, for instance providing information in regard to coverage of each sample (e.g. single cells from culture might give better coverage than primary CTCs, and might have very different features from pooled primary tumor cells).
- (3) Mutational analysis: how do the authors identify their mutations as cancer-associated? This is quite a strong claim, but from the tables presented many appear to be listed as SNPs instead.
- (4) It is unclear why the authors attempt to "force" their model into an EMT scale. Clearly, based on protein markers these cells appear to be epithelial. RNA data interpretation should be subordinate to protein analysis, therefore any statement like "shows partial EMT" (like in the abstract) is not supported by data, and seems unnecessary.
- (5) Same as above for CSC status assessment based exclusively on ALDH positivity. Per definition, CSCs should be able to self-renew, differentiate, and recreate a heterogeneous tumor from a single cell. None of these properties are tested/shown for the CTC-ITB-01 cells, and statements on their CSC content (like in the abstract) are not supported.
- (6) Fig 7C: differences in ER target genes in the CTC line are not visible.
- (7) In vitro data with CDK4/6 inhibitors should be complemented by in vivo data

Minor:

- (1) Fig. 7: when shRNAs are used, authors should refer to them as knockdown rather than knockout

EMM-2019-11908

***** Reviewer's comments *****

Referee #1 (Remarks for Author):

The manuscript is of good technical quality, although, in reviewer's opinion the most complete and well performed part is the characterization of the established CTC cell line (CTC-ITB-01) in vivo. Conversely, the characterization of CTC-ITB-01 cells in vitro in order to establish the grade of tumor heterogeneity in culture and the stemness is lacking in some aspects. It is not novel (see previous papers i.e. Aceto et al. Cell. 2014 Aug 28;158(5):1110-1122) but it provides an additional contribution to address the issue of Circulating Tumor Cells (CTCs) which is still largely unknown. In particular, it provides a novel model to study ER+ breast cancer CTCs Thus it may be of impact for further development of novel anti-cancer therapies. Therefore, Reviewer believes that the manuscript is suitable for publication after a minor revision.

Following a detailed analysis of the manuscript highlighting the main concerns that should be addressed.

In this manuscript authors provide an in-deep characterization of CTC-ITB-01 cells, a cell line established by isolation of CTCs from whole blood of a metastatic breast cancer patient. They show that CTC-ITB-01 cells show an epithelial ER+ positive phenotype and suggest that this cell line may reveal a novel suitable model to study ER+ breast cancer CTCs.

In this manuscript authors evaluate:

- morphology of cells
- copy number alterations (CAN),
- expression-based subtyping using the PAM50 and scmod2 classifiers.
- evaluation of Stem markers frequently showed by cancer cells endowed with high phenotypic plasticity. (CD44, CD24, ALDH, E-cadherin, alpha-catenin, EMT-scoring algorithm analyzing RNA-sequencing data).
- in vivo functional characterization of tumorigenic and metastatic potential.
- evaluation of endocrine sensitivity and chemoresistance.

1. The first and main concern regards the method used for isolation of CTCs. CTCs identification and count through CellSearch® technology is based on staining of an epithelial marker (EpCAM) allowing to distinguish CTCs as EpCAM+ cells from leukocytes which are EpCAM-. The signature based on EpCAM, pancytokeratins and CD45 makes the CellSearch® System a reliable instrument with prognostic value in metastatic cancer patients (breast, colon and prostate). However, as the authors say at page 10: "Since cell culture is not possible on the fixed CTC fraction from the CellSearch® cartridge, we took an additional, parallel blood sample from the same patient and cultured CTCs enriched by negative depletion of leukocytes which is unbiased for a particular phenotype of CTCs".

Reviewer believes it should be performed a cytofluorimetric analysis establishing the amount of EpCAM+, CD45+ and CD34+ cells upon the establishment of the CTC-ITB-01 cell line. This analysis could highlight if some endothelial progenitors may persist during culture of CTC-ITB-01 cells. This is also of particular importance in order to establish if VEGF identified by functional fluoro-EPISPOT assay (Supplementary Figure 6) is secreted by CTC-ITB-01 cells or, rather it is produced by endothelial progenitors.

Reply:

We agree in principle with the reviewer that this marker-independent enrichment method for CTCs offers the possibility also for non-tumoral CD45-negative cells to get enriched and grow. However, the endothelial progenitor cells are not able to survive a high number of passages over two years in an intact and unaffected state (Ishikawa & Asahara, *Stem Cells Dev.* 2004), and furthermore our genetic and molecular characterization studies clearly show that CTC-ITB-01 cells are tumor cells.

Moreover, we stained CTC-ITB-01 cells on slides (cytospins) with an anti CD34 antibody (stains endothelial progenitor cells) and did not find CD34-specific immunofluorescence (see Figure 1 below). Kasumi-1 (AML cell line, CD34-positive, Asou H et al. *Blood* (1991) 9:2031-2036) and MCF-7 breast cancer cells served as positive and negative control, respectively. Therefore, it is highly unlikely that VEGF is secreted by endothelial progenitor cells rather than by CTC-ITB-01 breast cancer cells.

Kasumi-1

CTC-ITB-01

MCF-7

Figure: Immunofluorescence staining of CTC-ITB cells with an anti-CD34-FITC-labeled antibody. Kasumi-1 and MCF-7 breast cancer cells served as positive and negative control, respectively.

2.1. The characterization of tumor heterogeneity due to expression of stem markers and EMT in CTC-ITB-01 cell line is not properly addressed. Authors stated that the "morphology of the CTC-ITB-01 cell line is heterogeneous and not typical for an epithelial-like breast cancer cell

line such as MCF-7" (at the end of Page 10). However, Figure 1C show well-shaped epithelial cells and this heterogeneity does not appear clearly.

In addition, at the end of page 10 authors say that cultivating both adherent and non-adherent cells separately "gives rise to the respective counterpart, indicating a high plasticity of CTC-ITB-01 cells similar to that recently reported for cancer stem cells [41]." However, in reviewer's opinion, all the stemness markers evaluated strongly suggest that CTC-ITB-01 cells have a well-defined epithelial phenotype: cortical E-cadherin and cortical beta-catenin localization, low expression of CD44, vimentin and N-cadherin, high expression of CD24, CK8-18 positivity associated to ER expression.

Also the PAM-50 classifier shows a great alignment of CTC-ITB-01 with a luminal B subtype. Thus heterogeneity based on morphology and epithelial-mesenchymal markers in CTC-ITB-01 seems to be lacking.

Reply:

The reviewer is right that Figure 1C shows adherent cells with epithelial appearance. However, Figure 1D also shows non-adherent floating cells. Moreover, in Figures 5A and 5B some adherent elongated cells with a more mesenchymal-like phenotype are to be seen while others have a rather epithelial morphology as evidence for heterogeneity. What is also shown on Fig 5A and 5B is that round cells have E-cadherin on the cell surface; E-cadherin was not found at cell-cell contacts since most cells are isolated. Furthermore, CTC-ITB-01 cells express CDH-1 molecules that are adhesion-deficient (mutation in EC3). Moreover, as Sarrió et al. showed also luminal tumors might contain subsets of carcinoma cells with EMT markers (Sarrió et al. *Cancer Res* 2008, 68:989-997) which is corroborated by the results obtained with our cell line. Besides that, CTC-ITB cells are characterized by a lower expression of epithelial keratins 8, and 18 in comparison to MCF-7 cells (Figure 4A) which is even considered a hallmark of EMT (Aside et al. *Cancer Res* 2014 71:4707-19, Shi et al. *Oncol Rep* 2019, 41:3015-26, Fortier AM et al. *J Biol Chem* 2013, 16:11555-11571, Willipinski-Stapelfeldt et al. *Clin Cancer Res* 2005, 11:8006-14.).

Furthermore, by Western blot analysis we could show that CTC-ITB-01 cell line cells express SLUG and higher levels of TWIST in comparison to MCF-7 cells, which is also indicative of EMT at least in a subset of cells (Figure 4A).

Additionally, we replaced the sentence "Cultivating of either fraction separately gives rise to the respective counterpart, indicating a high plasticity of CTC-ITB-01 cells similar to that recently reported for cancer stem cells [41]." by "Cultivating of either fraction separately results first in adherently growing cells and subsequently these cells give rise to the development of tumor cells in suspension, indicating a high plasticity of CTC-ITB-01 cells on page 10.

2.2. However, if authors want to evaluate stem-like properties of isolated CTC-ITB-01 cells they could try to obtain tumors after xenografts at limiting dilutions or evaluate activation of some stem pathways such as Notch, Numb, Hedgehog, excluding wnt signaling since cortical beta-catenin localization (Figure 5) suggests that it might be uninvolved. In addition, to establish if an Epithelial Mesenchymal Transition Programme (EMT) is activated in CTC-ITB-01 cells, authors could evaluate expression of some EMT marker as Twist, Snail1, Slug, c-Met. Even if this markers could be expressed in CTC-ITB-01 cells just because they are metastatic cells with high migratory activity however this information could be useful for those researchers interested in the use of this cell line model.

Reply:

We agree with the reviewer that we have to provide more results to confirm stem-like properties of CTC-ITB-01 cells. However, xenografts of single CTC-ITB-01 cells at limiting dilutions are not realistic in the given time frame and may be addressed in a follow-up publication. Nevertheless, to evaluate stem-like properties of CTC-ITB-01 cells we extended our Western blot analyses on cell lysates from CTC-ITB-01 cells in comparison to MCF-7 (ER+ epithelial phenotype) and Hs578t cells (ER- mesenchymal-like phenotype, Figure 4A and Figure EV5). Interestingly, CTC-ITB-01 cell line cells express high amounts of NUMB, indicating activation of the NUMB pathway which is associated with a deactivation of the NOTCH pathway and down-regulation of NOTCH1, NOTCH3 but not cleaved NOTCH1 (see Fig EV5xy). NUMB is known to rather function as an onco-suppressor by inactivation of NOTCH signaling (Colaluca et al. *J Biol Chem* 2018, 217:745-762), thereby preventing complete EMT and stabilizing an E/M hybrid phenotype. NUMB was also associated with high aggressiveness in lung and ovarian cancer (Bocci et al. *J. R. Soc. Interface* 2017, 14:20170512.)

As already mentioned above, our cell line expressed higher amounts of EMT transcription factors TWIST and SLUG as MCF-7, while SNAIL expression was lower in CTC-ITB-01 than in MCF-7 (see Figure 4A).

In conclusion, our results suggest that at least subsets of CTC-ITB-01 cell line cells have EMT-like properties and stem cell pathways are involved in maintaining epithelial features and stabilizing an E/M hybrid phenotype in these cells (now on pages 9 and 10).

Minor revision

1- In supplementary Figure 1 E-cadherin staining in Formalin-fixed paraffin-embedded (FFPE) tissue of both primary tumors was shown. Authors show exemplary IHC images of both the primary lobular and ductal tumors. They should provide a magnification of both picture allowing the identification of E-Cadherin localization in breast tissues, in addition they should provide also the staining in the normal breast tissues.

Reply:

In order to respond to the reviewer's comments, we prepared a new figure with higher magnification and also show E-cadherin immunostaining in normal appearing ductal epithelium (see Fig EV1).

2- If the existence of a mixed population showing mesenchymal or epithelial morphology is proven then Figure 2C should be replaced because the cells in the picture clearly appear as epithelial cells.

Reply:

We assume the reviewer means Figure 1C here, however, we do not agree to replace Figure 1C. We chose Figures 1C and 1D since they should only show the two different states of the cells, adherent and suspension cells. Moreover, Figure 1D already shows the round floating cells.

3- The following paragraph at page 10 is not clear for the non-specialists, in reviewer's opinion it may be misleading: "The high CTC count of the CellSearch® analysis (1,547 CTCs per ml of blood) indicated that a large number of the in situ CTCs originally present in the blood of the cancer patient expressed epithelial markers (EpCAM and keratins) [40]."

Reply:

We agree with the reviewer and adjusted the paragraph accordingly:

“In the initial blood sample of the breast cancer patient, the high number of more than 1000 CTCs per ml of blood was detected with the FDA-cleared CellSearch system used in our clinical studies (Bidard et al., *Lancet Oncology* 2014, 15:206-14; Riethdorf et al., *Clin Cancer Res* 2017, 13:920-8); CellSearch uses magnetic particles coupled to antibodies against EpCAM to enrich fixed CTCs and subsequently identifies single CTCs by immunostaining of epithelial keratins. Thus, we conclude that this patient harbored many CTCs with an epithelial phenotype. However, we could not exclude that mesenchymal CTC phenotypes lacking EpCAM or keratin expression were also present in the blood of this patient but remained undetected by CellSearch, as shown previously (Castro-Giner and Aceto, *Genome Med* 2020, 12:31). To avoid selection bias for a particular phenotype of CTCs [6], we therefore took another blood sample from the same patient and cultured CTCs that were enriched by depletion of leukocytes using the Rosette Sep technology; this negative selection approach allows an enrichment independent from the CTC phenotype (Pantel and Alix-Panabieres, *Nat Rev Clin Oncol*, 16:409-24). Thus, CTCs with epithelial and mesenchymal attributes had the same chance to be enriched and grown in culture. This may explain the documented phenotypic plasticity of our cell line cells (see reply above).

4- In the caption of Supplementary Figure 5 there is a mistake in point (2) ALH1 rather the ALDH1

Reply:

We thank the reviewer for pointing out this oversight. It has been corrected to ALDH1.

5- In the caption of Supplementary Figure 6 the meaning of T- e T+ is not described.

Reply:

We thank the reviewer for pointing out this oversight. This Figure is now Appendix Figure S2. We replaced T- and T+ by Pos. Ctrl. and Neg. Ctrl and explained it in the legend as follows: Pos. Ctrl.: Cell lines secreting the appropriate proteins, Neg. Ctrl. No cells added.

Referee #2 (Remarks for Author):

This manuscript reports the characterization of a new cell line obtained from circulating tumor cells of a breast cancer patient. It will constitute a very important tool to study tumor dissemination in luminal breast cancer. These results are quite remarkable as it is very difficult and rare to obtain cell line and PDX from estrogen positive breast cancers, even more from CTC and not tumor samples. The characterization of the cell line is very comprehensive, from whole genome sequencing and extensive immune cytology description to PDX formation. The results are well discussed by experts in the field.

Reply:

We thank the reviewer for this very positive evaluation of our work and the helpful comments.

Comments:

(1) As the probability of success to obtain a cell line from CTC is very low, it would be of interest to know how many patients were screened before the success with this patient?

Reply:

The reviewer addresses a valid point. We have included the number of metastatic breast cancer patients screened in this study as a point of reference. This information has been added to the Materials and Methods section under “Blood collection and processing”, page 18:

“For this study the blood of 50 metastatic breast cancer patients was collected and processed.”

(2) In the clinical presentation of the patient case, the percentage of estrogen and progesterone receptor positivity would be of interest to be provided in the text and not only in supplementary table.

Reply:

We agree with the reviewer and have added the information to the text in the “Results” section on page 5 “Patient characteristics”:

“Both tumors were classified as estrogen receptor positive (ER⁺ in ≥80% of nuclei), progesterone receptor positive (PR⁺ in ≥80% of nuclei), ERBB2 negative (ERBB2⁻), and presented with a low Ki67⁺ cell fraction of 5%.”

(3) P19, it is reported that genomic DNA from primary tumor and from the vaginal metastasis was used for genomic sequencing. However, there is no result reported of the vaginal metastasis in the genomic section of the results page 7. Was the vaginal metastasis biopsied and analyzed and was it compared to CTC and primary tumors profiles?

Reply:

We thank the reviewer for pointing out this oversight. We now also included and discuss the results for the vaginal metastasis.

The two primary tumors, the vaginal metastasis and the CTC cell line shared a mutation in the gene region encoding the kinase domain of the PIK3CA protein (c.3140A>G; p.H1047R, Table 1), a somatic hot spot mutation site in lobular and ductal breast cancer that has been associated with increased enzymatic activity of PIK3CA. Besides shared variations with the primary tumors, CTC-ITB-01 and the vaginal metastasis exhibited an additional, less frequent *PIK3CA* mutation (c.1252G>A; p.E418K, Table 1), located in the region encoding the C2 calcium/lipid-binding domain. Moreover, we identified a somatic genomic aberration of the *NFI* gene that was shared by the CTC-ITB-01 cell line, the left primary lobular tumor and the vaginal metastasis (Table 1). However, the vaginal metastasis carried also “private mutations” not observed in the primary tumors or the CTC line (e.g., *CDH1* gene mutation [c.2466delC; p.T823Qfs*23]) (Table 1). Taken together, we can speculate that the vaginal metastasis might have contributed to the pool of CTCs but it appears not be the exclusive source. We changed the text on pages 8 and 9, and 14 and 15 appropriately.

(4) It would be interesting to give the date of treatment of the patient, in order to understand which treatment were not available like CDKi in this time.

Reply:

The patient was treated between 2012 and 2014. At this point in time, CDKi were not available for standard of care in breast cancer in Germany (now stated on page 13). To our knowledge, the first CDKi (Palbociclib) was approved in the EU in 2016.

In order to respond to the reviewer's comment, we adjusted the manuscript on page 14.

(5) In the results, Table 1 about LOH, could be transferred into the EV tables (Table EV6) and a new table with a summary of the main clinical significant mutations detected by WGS extracted from the supplementary table would be of more interest.

Reply:

We agree with the reviewer and transferred this Table into the EV material as Table EV2. We now show a summary of clinically relevant cancer-associated mutations for the CTC-ITB-01 cell line, the primary tumors and the vaginal metastasis in Table 1. All four samples were analyzed for rare (MAF <1%), functionally relevant variants with a allele frequency of 10% and higher in at least one of the samples. 219 cancer-associated genes curated from the COSMIC, HGMD and OMIM databases (now Appendix Table 1) were analyzed and 19 variants in 13 genes were identified. Gene symbol were used as approved by the HGNC, and location of variants on cDNA and protein (one letter code) level, and allele frequency is shown in Table 1.

(6) In figure 2, CNA evaluation was performed on cell line culture, on primary tumor but also on CTC collected at the first blood draw. How were CTC counted to perform NGS on 1 or 5 or 10 CTC? Was it micropipetting or a specific device to perform NGS on isolated cells?

Reply:

Single CTCs were isolated using manual micromanipulation. We have added this information to the "Materials and Methods" section in the paragraph "Next generation sequencing and CAN profiling":

"Single immunostained CTCs or pools of CTCs were picked via micromanipulation using a fluorescence microscope and underwent whole genome amplification using the PicoPLEX DNA-seq kit followed by library preparation according to manufacturer's instructions (Takara Bio)" on page 19.

(7) Mutation of CDH1 was reported. Is there any result to hypothesize that this mutation is subclonal in order to explain the expression of E-cadherin by the tumor?

Subclonal or parallel

Reply:

We did not find this particular mutation in the primary tumors. However, whether this mutation is subclonal or has evolved in parallel cannot be concluded from our results. What is also shown on Fig 5A and 5B is that round cells have E-cadherin on the cell surface; E-cadherin was not found at cell-cell contacts since most cells are isolated. Furthermore, CTC-ITB-01 cells express CDH-1 molecules that are adhesion-deficient (mutation in EC3).

CDH1 staining is still observed in the CTC-ITB cell line despite the mutation but the subcellular distribution is disturbed.

(8) Page 13, could the authors clarify what is PR A and B.

Reply:

We have added the clarification to the respective sentence of the manuscript on page 13.

“This data suggests that ER-alpha within CTC-ITB-01 cells is active even in the absence of E2, which might explain why CTC-ITB-01 can tolerate low E2 levels. Interestingly, while MCF-7 expresses PR A (progesterone receptor α) and B (progesterone receptor β) in an E2-dependent manner, CTC-ITB-01 cells display low amounts of PR A but not B.”

(9) The authors have tested the impact of palbociclib, a CDK inhibitor, on the CTC-ITB1 cell line, which show some characteristics associated with resistance to endocrine treatment. The CDKi are particularly of interest if combined with endocrine treatment. This has not been tested. Is there a specific reason for this strategy of monotherapy and not a combination with tamoxifen or fulvestrant?

Reply:

This is correct, CDK inhibitors were tested as monotherapy on the cell line in this study. The reasoning behind this decision was that the patient had already received prior endocrine therapy, in form of aromatase inhibitors. Her disease had progressed under therapy before the cell line was established, indicating therapy resistance. In combination with results provided in our study that indicate the ER pathway remains constitutively active in the cell line, the remaining underlying question was whether CDK inhibitors could provide additional benefit.

(10) In the discussion or introduction about CTC clusters, the authors could quote recent work by Gkountela S et al in Cell 2019.

Reply:

We have included the reference in question in the introduction, page 3.

(11) In the discussion, there is no mention of ESR1 mutation as a frequent mechanism of resistance to anti hormonal treatment, more frequently when exposed to aromatase inhibitor.

Reply:

We thank the reviewer for this comment. We have added the following sentence and reference to the “Discussion” section of the manuscript: page 17.

“While ESR1 mutations represent a common mechanism of acquired resistance to endocrine therapy, we did not detect these mutations in CTC-ITB-01, suggesting a different mechanism of resistance in our index patient.”

Referee #3 (Comments on Novelty/Model System for Author):

Low novelty: breast CTC culture and xenografts have been achieved in the past by several other groups (e.g. Yu et al, *Science* 2014).

Reply:

According to recent Editorials in *Science* (Ma and Jeffrey, *Science* 2020, 367:1424-25) and *Nature* (Alix-Panabieres, *Nature* 2020, 579(7800):S9), there is still an unmet need to establish (and publish) more CTC cell line models because they open a new avenue for understanding CTC biology. Thus, we are confident that our present work adds important information to the ground-breaking publication by the team of Min Yu and Daniel Haber (Yu et al, *Science* 2014, 345:216-20) and others in this new field of research. The establishment of a panel of CTC lines from different patients (and tumor entities) will open new avenues to understand the biology of cancer metastasis. We also like to emphasize that our new CTC line is a permanent line that we could share with several international expert teams. Additionally to the study of Yu et al. we also genomically compared the primary tumors, original CTCs and the CTC-ITB-01 cell line concerning CNVs.

Referee #3 (Remarks for Author):

Koch, Kuske and colleagues describe the establishment and characterization of a novel CTC breast cancer cell line. They demonstrate that this ex vivo model retains ER positivity in culture and genomic (CNV) concordance with the original CTCs from the very same patient. They also suggest that the established CTC line presents with partial EMT and cancer stem cell features, and demonstrate the ability of these cells to form metastasis in a PDX model. Together, this manuscript provides an interesting new model to study CTC biology, but some considerations should be made:

- (1) Breast CTC culture and xenografts have been shown in the past, including ER+ models with metastatic ability in animal models (e.g., Yu et al., *Science* 2014, 345:216-20 among others). From the current submission it is unclear what is the major novelty of this paper, compared to previous studies.

Reply:

According to recent review articles (Keller and Pantel, *Nat Rev Cancer* 2019, 19:553-567) and Editorials in *Science* (Ma and Jeffrey, *Science* 2020, 367:1424-25) and *Nature* (Alix-Panabieres, *Nature* 2020, 579(7800):S9), there is still an unmet need to establish (and publish) more CTC cell line models because they open a new avenue for understanding CTC biology. Thus, we are confident that our present work adds important information to the ground-breaking publication by the team of Min Yu and Daniel Haber (Yu et al, *Science* 2014) and others in this new field of research. We also like to emphasize that our new CTC line is a permanent line that we could already share with several international expert teams. Additionally to the study of Yu et al. we also compared the primary tumors, distant metastasis, original CTCs and the CTC-ITB-01 cell line at the genomic level (CNVs).

- (2) The authors interpret the CNV results (Figure 2) as "tumor evolution during the course of the disease", "the CTC line derived from a subpopulation of CTCs", and "CTC line stability" depending on specific comparisons. However, they should exclude potential technical biases

of this type of analysis, for instance providing information in regard to coverage of each sample (e.g. single cells from culture might give better coverage than primary CTCs, and might have very different features from pooled primary tumor cells).

Reply:

To avoid the biases mentioned by the reviewer, we selected only those primary CTCs that had an intact morphology and we pre-tested the DNA quality before CNA evaluation to make sure that the quality of DNA is as good as for the cell pools or cell line cells.

In order to respond to the reviewer's comment, we calculated the mean sequencing depth over the whole genome for the primary CTCs (mean, 0.0781; std, 0.0326) and the cell line cells (mean, 0.1014; std, 0.0291); however, there is no statistical significant difference between the two ($p=0.089$, Welch's two sample t-test). Because the tumor tissue was sequenced by WES, the mean sequencing depth over the whole genome was of course more (mean, 1.951; std, 0.81), however, to be able to compare all experiments with each other and to minimize any technical bias, the resolution of all copy number data was downscaled to 500 kB.

(3) Mutational analysis: how do the authors identify their mutations as cancer-associated? This is quite a strong claim, but from the tables presented many appear to be listed as SNPs instead.

Reply:

We thank the reviewer for his comment. We agree that *not all* changes measured are tumor-derived but the key goal of this genomic evaluation was to compare primary CTC and CTC cell line cells and make sure that our cell line is CTC-derived.

Therefore we replaced the Tables by Table 1 and Appendix Table 1. All four samples (CTC-ITB-01 cell line, both primary tumors and the vaginal metastasis) were analyzed for rare (MAF <1%), functionally relevant variants with a allele frequency of 10% and higher in at least one of the samples. 219 cancer-associated genes curated from the COSMIC, HGMD and OMIM databases (now Appendix Table 1) were analyzed and 19 variants in 13 genes were identified. Gene symbol were used as approved by the HGNC, and location of variants on cDNA and protein (one letter code) level, and allele frequency is shown. Thus, we have demonstrated copy number aberrations that are beyond SNPs and clearly linked to cancer development and progression such as those in *PIK3CA* and *TP53*. Therefore, we are confident that our CTC line cells are tumor-derived.

(4) It is unclear why the authors attempt to "force" their model into an EMT scale. Clearly, based on protein markers these cells appear to be epithelial. RNA data interpretation should be subordinate to protein analysis, therefore any statement like "shows partial EMT" (like in the abstract) is not supported by data, and seems unnecessary.

Reply:

As already mentioned above (answer to reviewer 1), our cell line expressed higher amounts of EMT transcription factors TWIST and SLUG as for example MCF-7 breast cancer cell line cells.

In conclusion, our results suggest that subsets of CTC-ITB-01 cell line cells have EMT-like properties and stem cell pathways are involved in maintaining epithelial features and stabilizing an E/M hybrid phenotype in these cells. We appropriately modified the abstract and text of the revised manuscript and added a comment in the Discussion section pages 15 and 16.

(5) Same as above for CSC status assessment based exclusively on ALDH positivity. Per definition, CSCs should be able to self-renew, differentiate, and recreate a heterogeneous tumor from a single cell. None of these properties are tested/shown for the CTC-ITB-01 cells, and statements on their CSC content (like in the abstract) are not supported.

Reply:

We agree with the reviewer that the definition of cancer stem cells. However, the CSC field has become very complex and our CTC line shows several features that have been called “CSC-associated” in the literature, including EMT-like features or ALDH1-positivity. Nevertheless, we fully respect the comment of the reviewer and have modified the wording in the Abstract and text of the revised manuscript and added a comment in the Discussion section page 15.

See also answer to Reviewer 1

To evaluate stem-like properties of CTC-ITB-01 cells we extended our Western blot analyses on cell lysates from CTC-ITB-01 cells in comparison to MCF-7 (ER+ epithelial phenotype) and Hs578t cells (ER- mesenchymal-like phenotype, Figure 4A).

Interestingly, CTC-ITB-01 cell line cells express high amounts of NUMB, indicating activation of the NUMB pathway which is associated with a deactivation of the NOTCH pathway and down-regulation of NOTCH1, NOTCH3 but not cleaved NOTCH1 (see Figure EV5). NUMB is known to rather function as an onco-suppressor by inactivation of NOTCH signaling (Colaluca et al. *I Biol Chem* 2018, 217:745-762), thereby preventing complete EMT and stabilizing an E/M hybrid phenotype. NUMB was also associated with high aggressiveness in lung and ovarian cancer (Bocci et al. *J. R. Soc. Interface* 2017, 14:20170512.)

(6) Fig 7C: differences in ER target genes in the CTC line are not visible.

Reply:

We thank the reviewer for this comment. CTC-ITB-01 cells indeed show no differences in ER target genes. Therefore, we deleted the sentence “CTC and MCF7 cells show differences in the expression of ER target genes”.

In Figure 7C differences in protein level between MCF7 and CTC-ITB-01 cells that are treated and untreated with 17 β -estradiol are demonstrated. These differences in protein abundance are most pronounced for FOXM1 and both subunits of the progesterone receptor but are also clearly visible for Bcl-2 and Id1.

(7) In vitro data with CDK4/6 inhibitors should be complemented by in vivo data

Reply:

Single therapy of CTCs derived from breast cancer patients with CDK inhibitors such as Ribociclib and Palbociclib has been shown in the past by Yu *et al.* (*Science* 345, 216-220). While their breast cancer CTCs did not respond well to treatment, our CTC line ITB-01 was sensitive to the CDK inhibitor. We included these data from the *in vitro* experiments to underline the heterogeneity of CTCs obtained from breast cancer patients, which further supports our initial response (see above) that more CTC lines need to be established to fully understand the full spectrum of CTC biology in breast cancer and other tumors. The *in vitro* experiments in our study were primarily conducted to show the principle response of CTCs to CDKi but we fully understand that much more *in vivo* pre-clinical work is required in future studies, which is far beyond the scope of the present paper. Here, through the drug sensitivity assay we just wanted to emphasize that drug screening on CTCs is achievable, which may open up new therapy strategies for extensive *in vivo* work. In addition, our results from several studies resulted in similar effects of drugs tested *in vitro* and *in vivo* in xenotransplantation models (Grabinski *et al.*, *Mol Cancer* 2012, 11:85, Ewald *et al.*, *Int J Cancer* 2013, 133:2065-76) and we are therefore confident that our current *in vitro* findings point into a fruitful direction.

Nevertheless, we have added a comment of caution to the section in the Discussion that describes the limitations of the current manuscript and the need of future studies.

Minor:

(1) Fig. 7: when shRNAs are used, authors should refer to them as knockdown rather than knockout

Reply:

We thank the reviewer for this comment and have adjusted Figure 7 accordingly.

27th May 2020

Dear Dr. Pantel,

Thank you for the submission of your revised manuscript to EMBO Molecular Medicine, and please accept my apologies for the delay in getting back to you in these unusual times. We have now received the enclosed reports from the two referees who reviewed the new version of your manuscript. As you will see, while referee #1 is now supportive of publication, referee #3 still has a few concerns that should be addressed adequately. At this stage, we would like you to discuss the referee's points in writing, and if you do have data at hand, we would be happy for you to include it, however we will not ask you to provide any additional experiments.

Furthermore, please address the following editorial amendments:

1) Main manuscript text:

- Please correct/answer the track changes suggested by our data editors in the main manuscript file (in track changes mode). I will send you the data-edited file shortly; please use this version of the manuscript for any further changes. Please also include a point-by-point rebuttal to referee #3's concerns.
- Remove the highlighted text.
- The references for Fig. 7D, E are missing in the manuscript text, please update.
- Your text was cross-checked for similarities with other manuscripts, and a resemblance was found with a previously published text. Please modify your introduction accordingly (see the parts of your text highlighted in the attached document).
- Author contribution: please differentiate between the contributions of Kerstin Borgmann and Kai Bartkowiak.
- In the Material and Method section: In line with our transparent policy, we aim at having most of the material and methods easily accessible in the main manuscript. Would you thus move (part of) the supplemental material and methods from the Appendix to the main manuscript? Along these lines, please make sure that all experimental methods are sufficiently described to be easily reproducible (i.e. limit the references to previously published methods, and make sure that these are published in open access articles).
- Please include the full sentence: "informed consent was obtained from all subjects and that the experiments conformed to the principles set out in the WMA Declaration of Helsinki and the Department of Health and Human Services Belmont Report."
- Please indicate in the figure or figure legends the exact $n=$ and exact $p=$ values, not a range, along with the statistical test used. Some people found that to keep the figures clear, providing a supplemental table with all exact p -values was preferable. You are welcome to do this if you want to.
- Data availability: thank you for depositing your data at the European Nucleotide Archive, however I could not access the data with the accession number PRJEB37968. This data has to be made accessible before acceptance of the manuscript. The accession number does not need to be further cited in the material and methods section.

2) Checklist:

- In part B-statistics: please make sure that all fields are filled adequately.
- In part C-reagents: please update part 6 (antibodies).
- In part F-data accessibility: please fill in the section 20, access to human genomic datasets.

3) Figures:

- Fig. 1B is missing scale bars.
- Fig. 4C contains a typo "Clinical samples"
- Fig. 5A: the text is not readable
- Fig. 7 has excessive white space.
- Appendix file: please change the nomenclature of the table, which should be "Appendix Table S1"
- Dataset EV legends: The 3 EV tables should be uploaded as separate table files.

4) Source data:

Thank you for providing source data. Could you please upload them as one file per main figure?

5) Synopsis:

Thank you for providing a nice synopsis image. However, when resized to our format (550px-wide), the text is very difficult to read. Could you please modify accordingly?

6) As part of the EMBO Publications transparent editorial process initiative (see our Editorial at <http://embomolmed.embopress.org/content/2/9/329>), EMBO Molecular Medicine will publish online a Review Process File (RPF) to accompany accepted manuscripts.

In the event of acceptance, this file will be published in conjunction with your paper and will include the anonymous referee reports, your point-by-point response and all pertinent correspondence relating to the manuscript. Let us know whether you agree with the publication of the RPF and as here, IF YOU WANT TO REMOVE OR NOT any figures from it prior to publication.

I look forward to reading a new revised version of your manuscript as soon as possible.

Yours sincerely,

Lise Roth

Lise Roth, Ph.D
Editor
EMBO Molecular Medicine

The system will prompt you to fill in your funding and payment information. This will allow Wiley to send you a quote for the article processing charge (APC) in case of acceptance. This quote takes into account any reduction or fee waivers that you may be eligible for. Authors do not need to pay any fees before their manuscript is accepted and transferred to our publisher.

***** Reviewer's comments *****

Referee #1 (Remarks for Author):

The manuscript is of good technical quality. It is not novel (see previous papers i.e. Aceto et al. Cell. 2014 Aug 28;158(5):1110-1122) but it provides an additional contribution to address the issue of Circulating Tumor Cells (CTCs) which is still largely unknown.

Authors addressed reviewer's criticism appropriately. The additional results and their discussion have significantly improved the manuscript.

Therefore, in reviewer's opinion, the manuscript is suitable for publication.

Referee #3 (Remarks for Author):

I appreciate the work done by Koch, Kuske and colleagues to reply to Reviewers' concern. I think the manuscript is certainly improved. Some weaknesses still remain however, which I provide below with the intent to give the authors some constructive criticism and to help the editors with their final decision.

(1) Novelty: I have no doubt that additional CTC-derived models are needed to better understand CTC biology, and fully agree with the authors on this. On the other hand, is the establishment and characterization of one single additional model enough to grant publication in this journal? How representative is this single model of the complexity of human breast cancer? It would be interesting to know why the cultivation of CTCs from this particular patient was successful while others were not, does it have to do with the initial high number of CTCs isolated?

(2) Conclusions on EMT (and plasticity/cancer stemness): in my opinion the authors are doing a very unnecessary effort to fit their model on a EMT/plasticity/cancer stemness scale. Floating cells and higher expression of TWIST/SLUG/SNAIL compared to MCF7 or NUMB pathway expression are not per se an indication of EMT nor plasticity nor stemness. For instance, the RNA seq data in Fig5 confirms in a pretty direct way the epithelial nature of both adherent and floating populations. I suggest the authors to refrain from claiming EMT/cancer stemness nature of these cells throughout the manuscript without having performed the proper assays (e.g. serial dilution transplantation, symmetric vs asymmetric cell division etc)

(3) Cell line characterization: given that it is only one cell line, I would urge the authors to present the mutation data with more clarity. I am fully convinced these are cancer cells, however, what are the DRIVER mutations? This is obviously a very different concept from "variants". In table 1, how many of these alterations are "cancer-associated mutations in breast cancer" (eg. PIK3CA E542K, E545K, H1047R, for instance) vs "random variations in cancer-associated genes"? How many of these alterations are somatic, i.e. NOT found in non-neoplastic cells of the patient, such as blood cells?

The authors performed the requested editorial changes.

Reviewer's comments

Referee #3 (Remarks for Author): I appreciate the work done by Koch, Kuske and colleagues to reply to Reviewers' concern. I think the manuscript is certainly improved. Some weaknesses still remain however, which I provide below with the intent to give the authors some constructive criticism and to help the editors with their final decision.

(1) Novelty: I have no doubt that additional CTC-derived models are needed to better understand CTC biology, and fully agree with the authors on this. On the other hand, is the establishment and characterization of one single additional model enough to grant publication in this journal? How representative is this single model of the complexity of human breast cancer? It would be interesting to know why the cultivation of CTCs from this particular patient was successful while others were not, does it have to do with the initial high number of CTCs isolated?

Reply: We are just starting to unravel the complex biology of CTCs, and permanent cell lines are useful models that are still rare. We understand the limitation of our work and a single cell line in general, but we would like to reply that the (breast) cancer field has made remarkable progress over the past decades using single cell lines like MCF7 or MDA-MB-231 in numerous studies. We agree that single cell lines can never encompass the entire heterogeneity of CTCs (Keller & Pantel, Nature Cancer Rev. 2019), but they are very attractive models that are well accepted by the research community. The in-depth characterization of a CTC line from one prostate cancer patient was just published in Nature Commun. (by the group of Françoise Farace from IGR in Paris, Faugeroux, V et al. Nature Commun 2020, 11:1884) and we therefore think that the in-depth characterization in EMBO Mol. Med will attract the attention of the readers of this excellent journal.

We have already discussed that the high number of CTCs isolated is certainly one reason why we were able to establish our CTC line. However, from our experience and the experience of our collaborators in our European networks we know that high CTC number is no guarantee to obtain a permanent cell line. Thus, we assume that the cells of our cell line have also the right biological properties of the most “aggressive” subset of CTCs. Future studies comparing CTC lines with CTCs that will stop proliferating after short term culture might help to define the molecular make up required for permanent survival and growth of CTCs. We have made a respective comment into the revised Discussion, page 14.

(2) Conclusions on EMT (and plasticity/cancer stemness): in my opinion the authors are doing a very unnecessary effort to fit their model on a EMT/plasticity/cancer stemness scale. Floating cells and higher expression of TWIST/SLUG/SNAIL compared to MCF7 or NUMB pathway expression are not per se an indication of EMT nor plasticity nor stemness. For instance, the RNA seq data in Fig5 confirms in a pretty direct way the epithelial nature of both adherent and floating populations. I suggest the authors to refrain from claiming EMT/cancer stemness nature of these cells throughout the manuscript without having performed the proper assays (e.g. serial dilution transplantation, symmetric vs asymmetric cell division etc)

Reply: We appreciate this comment and have revised our manuscript to avoid overstatements on the EMT and cancer stemness nature of our CTC line. We also added a short comment to our revised Discussion: “Our RNA sequencing data show that the CTC line is largely of an epithelial nature, which is consistent with the recent report of Dan Haber’s team in Science (Ebright RY et al. Science 2020), indicating that CTCs with an epithelial signature were the most aggressive subset. Although our CTC line shows some signs of cancer stem cells, more detailed future investigations are required to further determine whether our CTC line fulfils all criteria of real cancer stem cells.” on page 16.

(3) Cell line characterization: given that it is only one cell line, I would urge the authors to present the mutation data with more clarity. I am fully convinced these are cancer cells, however, what are the DRIVER mutations? This is obviously a very different concept from "variants". In table 1, how many of these alterations are "cancer-associated mutations in breast cancer" (eg. PIK3CA E542K, E545K, H1047R, for instance) vs "random variations in cancer-associated genes"? How many of these alterations are somatic, i.e. NOT found in non-neoplastic cells of the patient, such as blood cells?

Reply: We thank the reviewer for his remarks. The alterations, which we presented in Table 1, do not include any SNPs or polymorphism within the genes analyzed.

Still, for some of the shown alterations there is no functional data available confirming pathogenicity, and classification of these alterations relies only on *in silico* prediction, and annotation and frequencies in human population and cancer-related databases (Tate JG et al. *Nucleic Acids Res* 2019, 47:D941-D947, Karczewski KJ et al. *Nature* 2020, 581:434-443). To present the genetic data in a clear and concise way, we therefore divided table 1 in two parts.

In the first part (now new Table 1), we present all identified mutations that are known to be pathogenic. These mutations were specifically identified in cancer cells present in tissue samples from cancer patients and functionally characterized as important for tumor development or progression, including mutations in *TP53*, missense mutations in *PIK3CA* and loss-of-function mutations in *NF1*, *CDH1* and *MAP3K1*.

In the second part of table (now new Table EV 2), we show alterations of “cancer-associated” genes, for which a causal relationship to tumor development or progression is so far unclear. All of these variants, mainly missense variants, are predicted to have a damaging effect on protein function (based on different *in silico* prediction programs) and/or have been annotated in mutational databases, like e.g. the *Human Gene Mutational Database* (HGMD). Still, final functional evidence for their pathogenicity is currently missing, which is why these alterations were classified as “variants of unknown significance” (VUS).

To further improve the presentation of the genetic data in the main text and the table legend, we modified and replaced the expression “pathogenic variant” by mutation whenever dealing with clearly pathogenic, cancer-causing alterations.

Due to the lack of DNA of non-neoplastic cells from our patients (there is a high number of tumor cells also in peripheral blood), we were not able to discriminate between germline and somatic alterations.

Corresponding Author Name: Klaus Pantel

Journal Submitted to: EMBO Mol Med

Manuscript Number: EMM-2019-11908